# Determination of NO$_x$ emission rates of inland ships from on-shore measurements

Kai Krause[1], Folkard Wittrock[1], Andreas Richter[1], Dieter Busch[2], Anton Bergen[2], John P. Burrows[1], Steffen Freitag[2], and Olesia Halbherr[2]

[1]Institute of Environmental Physics, University of Bremen, Bremen, Germany
[2]State Agency for Nature, Environment and Consumer Protection in North Rhine-Westphalia (LANUV NRW), Recklinghausen, Germany

**Correspondence:** Kai Krause (kakrau@iup.physik.uni-bremen.de)

**Abstract.** Inland ships are an important source of NO$_x$, especially for cities along busy waterways. The amount and effect of such emissions depends on traffic density and NO$_x$ emission rates of individual vessels. Ship emission rates are typically derived using in-situ land measurements in relation to NO$_x$ emission factors, e.g. the amount of pollutants emitted by ships per amount of burnt fuel. In this study a different approach is taken and NO$_x$ emission rates are obtained in $\mathrm{g\,s^{-1}}$. Within the EU Life project Clean Inland Shipping (CLINSH), a new approach to calculate NO$_x$ emission rates from data of in situ measurement stations has been developed and is presented in this study. Peaks (i.e. elevated concentrations) of NO$_x$ were assigned to the corresponding source ships, using the AIS (automated identification system) signals they transmit. Each ship passage was simulated using a Gaussian-puff-model in order to derive the emission rate of the respective source ship. In total over 32900 ship passages have been monitored over the course of 4 years. The emission rates of NO$_x$ were investigated with respect to ship speed, ship size and direction of travel. Comparisons of the on-shore derived emission rates and those from on-board for selected CLINSH ships show good agreement. The derived emission rates are of similar magnitude as emission factors from previous studies. Most ships comply with existing limits due to grandfathering. The emission rates (in grams per second) can be directly used to investigate the effect of ship traffic on air quality, as the absolute emitted amount of pollutants per unit time is known. In contrast, for relative emission factors (in grams per kilogram fuel), further knowledge about the fuel consumption of the individual ships is needed, to calculate the amount of pollutants emitted per unit time.

## 1 Introduction

In cities along busy waterways such as the Rhine, the diesel engines of inland vessels are a significant source of emissions of pollutants (i.e. oxides of nitrogen (NO$_x$ = NO+NO$_2$), carbon monoxide (CO), and aerosols). The total amount and effect of these emissions depends on the traffic density along those waterways and the emissions of the individual vessels. In order to limit the effects of these emissions on air quality, the Central Commission for Navigation on the Rhine (CCNR) and the EU have established, during past decades, regulations for ship engines (EUD, 1998; EUR, 2016; CCNR, 2020). However, these regulations only apply to new engines (new ship construction or replacement of old engines). Engines on ships already in service are subject to grandfathering and do not have to comply with newer regulations. The effect of these requirements is

therefore limited, as ship engines have a long service life. There is no provision for continuous monitoring of emissions from ships in service, as is the case with road vehicles, for example. To determine the emissions of ship traffic, there has been a lack of measurements of both the ship traffic and the emissions from the different types of ship engines during real cruising operation. Consequently, a large number of assumptions had to be made in order to determine the mean emissions caused by the ships.

Shipping emission have been typically investigated using in situ instruments, either on-board or on-shore (e.g., Moldanová et al., 2009; Alföldy et al., 2013; Diesch et al., 2013; Beecken et al., 2014; Pirjola et al., 2014; Beecken et al., 2015; Kattner et al., 2015; Kurtenbach et al., 2016; Kattner, 2019; Ausmeel et al., 2019; Celik et al., 2020; Walden et al., 2021). Additionally, remote sensing techniques such as differential optical absorption spectroscopy (DOAS) (e.g., Berg et al., 2012; Seyler et al., 2017, 2019; Cheng et al., 2019; Krause et al., 2021) and more recently UAVs are used to investigate ship emissions (Zhou et al., 2019, 2020). In most studies, ship emissions are investigated only for short time periods or specifics ships. Usually, emission factors either in $\mathrm{g\,kg^{-1}}$ fuel or $\mathrm{g\,kWh^{-1}}$ are derived, but some studies also derive emission rates, e.g. in $\mathrm{kg\,h^{-1}}$. $NO_x$ emission factors $\mathrm{g\,kg^{-1}}$ or $\mathrm{g\,kWh^{-1}}$, can be derived from simultaneous $NO_x$ and $CO_2$ measurements. To convert emission factors to emission rates, additional knowledge about the fuel consumption of the individual vessels would be needed. However, emission rates can be derived directly and are usually reported by studies using remote sensing techniques. For example, Berg et al. (2012) showed the capability of airborne DOAS measurements to derive $NO_2$ and $SO_2$ emission rates of sea ships. But also other remote sensing techniques such as LIDAR (Berkhout et al., 2012), or UV cameras (Prata, 2014) can be used to derive $SO_2$ emission rates for sea ships. In general there is a lack of long time observations of emission factors or emission rates, especially for inland ships. The long term impact of shipping emissions has been investigated by modelling studies (Eyring et al., 2005; Ramacher et al., 2018, 2020; Tang et al., 2020; Wang et al., 2021; Jiang et al., 2021).

Within the EU Life Project Clean Inland Shipping (CLINSH), two methods to measure ship emissions were used. In-situ instruments on-board of ships were deployed to measure the emissions of the engines directly at the exhaust. Measurements were carried out on 40 inland vessels, which participated in the project. Absolute $NO_x$ emission rates (in $\mathrm{g\,s^{-1}}$) have been derived from these measurements. In addition, a method to derive absolute $NO_x$ emission rates from on-shore measurements of passing ships has been developed and is presented in this study. The retrieval concept builds on the approach presented in Krause et al. (2021), but the method has been improved and the algorithm can now be used with data measured by any standardized in-situ measurement station located in the vicinity of a river.

In total more than 32900 ship passages have been identified and analysed between 2017 and 2021 and provide a data set, which will be used in the future update of the inland waterway vessel emission register of the state of North Rhine-Westphalia. Generally, the $NO_x$ emission rates reflect the real driving conditions at this part of the Rhine. The derived $NO_x$ emission rates may be used directly as input for models, that describes the emission of ships and differentiate between ship sizes and speeds over ground. Similarly, the derived $NO_x$ emission rates can be used to build a ship emission inventory for the lower Rhine area. In contrast to more regularly reported emission factors in $\mathrm{g\,kg^{-1}}$ or $\mathrm{g\,kWh^{-1}}$, the derived $NO_x$ emission rates may be used directly without further assumptions regarding the fuel consumption of the ships, as the fuel consumption is already accounted

for in the $NO_x$ emission rates. At the same time, the $NO_x$ emission rates are only strictly correct for this specific part of the Rhine and can't be easily adapted to other rivers.

## 2 Measurement sites

For the CLINSH project, the State Agency for Nature, Environment and Consumer Protection in North Rhine-Westphalia (LANUV NRW) set up continuous measurement stations in Duisburg at the Duisburg Rhine Harbour (DURH) and in Neuss at the Neuss Rhine Harbour (NERH), which measure $NO_x$ concentration and meteorological parameters such as atmospheric pressure, humidity, temperature, wind speed and wind direction close to the river Rhine.

### 2.1 Instrumentation

Instrumentation at both measurement sites along the river Rhine, Duisburg Rhine Harbour (DURH) and Neuss Rhine Harbour (NERH), was identical (for specifications see Table 1). Nitrogen oxides were measured with an AC32M from Environnement S.A. (ENVEA) 3.5 m above ground, while meteorological parameters were obtained with a weather station from Lambrecht Meteo GmbH during the course of the campaign. The weather sensor measured wind speed (U) with a rotary anemometer and wind direction ($\theta$) with a wind vane at 10 m above ground. The time resolution for both measurement types is 0.2 Hz or 5 seconds.

The measurement principle to obtain NO and $NO_2$ is based on the emission of light during the chemical reaction between NO and ozone in the reaction chamber of the instrument. This reaction is called chemiluminescence and corresponds to oxidation of a NO molecule by ozone to an excited state ($NO_2^*$, R1). During the decay to its electronic ground state, the molecule emits light in a spectrum from 600 to 1200 nm (R2), which is measured with a photomultiplier. Since each molecule emits a defined amount of light, the measured signal is proportional to the sum of NO molecules in the air sample. $NO_2$ concentration in the air sample is determined indirectly in a second step by converting it to NO in a hot molybdenum converter (R3) and subsequently oxidizing it with ozone as described above. This yields $NO_x$ from which the concentration of $NO_2$ is obtained by subtracting the previously measured concentration of NO. These measurements correspond to the standard reference method specified in the DIN EN 14211. Molybdenum converters have known cross sensitivities to other oxidized atmospheric odd-nitrogen species (e.g. $HNO_2$, $HNO_3$, HONO), which can lead to overestimation of $NO_2$ and $NO_x$ levels.

$$NO + O_3 \rightarrow NO_2^* + O_2 \tag{R1}$$

$$NO_2^* \rightarrow NO_2 + h\nu \tag{R2}$$

$$2NO_2 \rightarrow 2NO + O_2 \tag{R3}$$

**Table 1.** Specifications of the used instruments.

| Measurements at DURH and NERH | Specifications |
| --- | --- |
| Nitrogen oxides | $NO_x$ ($NO+NO_2$) |
| Instrument | AC32M |
| Manufacturer | Environment S.A. (ENVEA) |
| Measurement principle | chemiluminescence |
| Measurement range | NO: $0-1200\ \mu g\,m^{-3}$ |
| | $NO_2$: $0-500\ \mu g\,m^{-3}$ |
| Measurement accuracy | $15\ \%$ |
| Air sampling height | 3.5 m above ground level |
| Meteorological parameters | wind speed (U), wind direction ($\theta$) |
| Instrument | EOLOS-IND static weather sensor |
| Manufacturer | Lambrecht Meteo GmbH |
| Measurement range | $\theta$: $0-360\ °$ |
| | U: $0.1-85\ m\,s^{-1}$ |
| Measurement accuracy | $\theta$: $\pm\,3\ °$ |
| | U: $\pm\,0.5\ m\,s^{-1}$ |
| Measurement height | 10 m above ground level |

Additionally, the measurement stations are equipped with AIS (automatic identification system) receivers, which deliver information on the passing ships. Under favourable wind conditions (wind blowing ship plumes towards the in situ systems), both measurement stations show strong enhancements of $NO_x$ when ships pass the measurement site, which can be clearly seen as a peak in the time series. Different views of the measurement sites are shown in Figure 1.

## 2.2  Duisburg Rhine Harbour (DURH)

In Duisburg, the measurement site is located on the eastern riverbank of the Rhine (51.460721 °N; 6.727486 °E, 28 m AMSL). As the predominant wind direction has a westerly component, emissions from ships are transported towards the measurement site for the majority of the time. Consequently a large number of pollution peaks from ships passing are identified in the measured $NO_x$ concentration time series (e.g. Figure 2). Generally, this measurement site is well located to derive $NO_x$ emission rates from ships sailing along the Rhine, because it is close to the Rhine and the entrance to the DURH harbour basins. Con-
sequently, the measured concentration peaks can be differentiated for ships that pass the measurement site in different driving conditions e.g., ships that drive upstream against the river current or downstream with the river current. This measurement station has been set up in October 2017 and is still active at the time of writing. In this study, measurements from 2017 until the end of 2021 are evaluated.

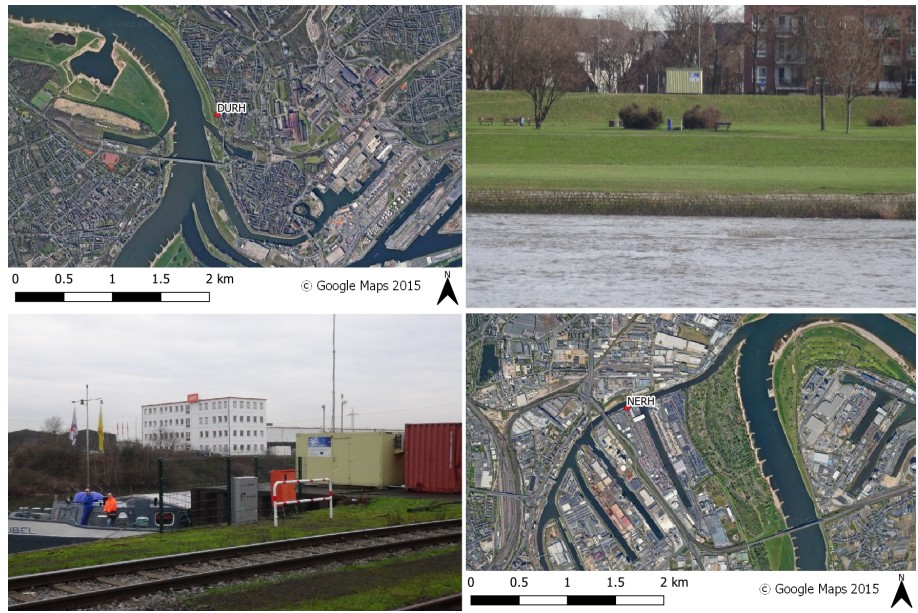

**Figure 1.** Views of the two different measurement sites used in this study. The upper row shows a satellite image of the DURH station and a picture of the measurement container as seen from the Rhine. The lower row shows a picture of the measurement container in the NERH and a satellite picture of its location.

## 2.3 Neuss Rhine Harbour (NERH)

In contrast to the measurement site DURH, the measurement site in Neuss is located within the Neuss harbour area on the west side of the Rhine (51.219577 °N; 6.704074 °E, 30 m AMSL). Buildings and vegetation block the direct line of sight from the measurement station to the Rhine. The combination of its location and the predominant south-westerly wind direction leads to only a few plumes being detected at this measurement site from ships that are steaming along the Rhine. Nevertheless, due to its location directly within the harbour, this measurement site is well suited to evaluate the emissions of slow moving ships within the harbour area where the influence of the river currents on engine operation are negligible. This measurement station was set up in September 2017 and dismantled at the end of 2019. Therefore, $NO_x$ emission rates could be derived for the years 2017 to 2019.

## 2.4 On-board measurements

## 3 Methods

Combining the onshore in-situ measurements of $NO_x$ and the received AIS signals enables ship emission rates from passing ships, identified by AIS, to be calculated. The approach uses three consecutive steps, which are described in the following.

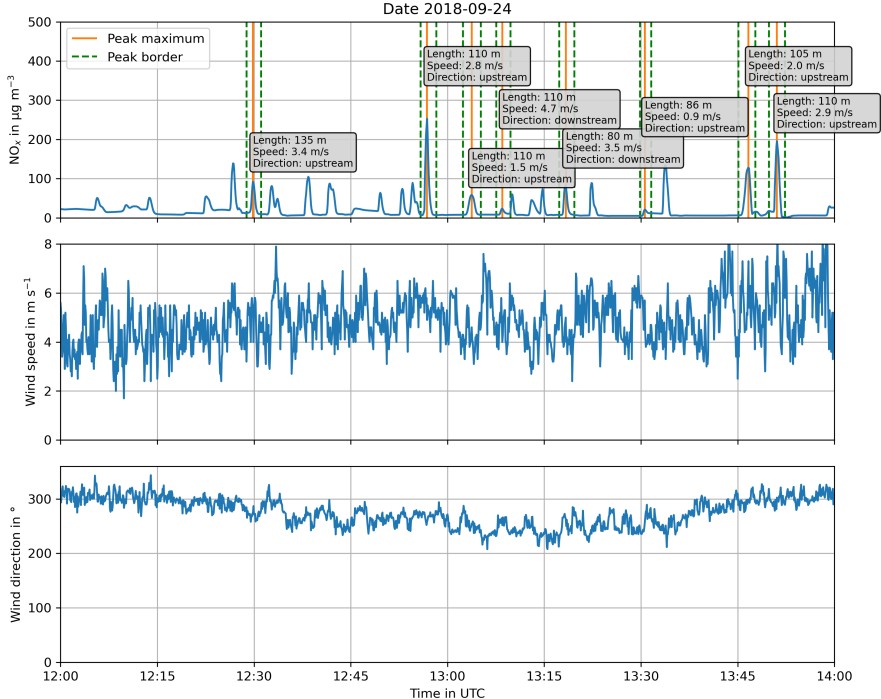

**Figure 2.** Example of the measured $NO_x$ concentration, wind speed and wind direction at DURH (5 s time resolution). Ship peaks identified in the $NO_x$ concentration are marked with an orange line, their borders are green dashed lines. The text box at each peak shows the ship length in m, the speed over ground in $m\,s^{-1}$ and the direction of travel (upstream or downstream). Peaks without a label are most likely also caused by passing ships, but in these cases, unambiguous assignment of a source was not possible.

## 3.1 Peak identification

The first step is to identify the peaks of $NO_x$ caused by passing ships. To identify these peaks, a low pass filtered time series is calculated from the measured time series using a running median with a window length of 5 minutes. This low pass time
series describes the changes in the background concentration caused by meteorological factors and other emission sources, but excludes the short-term variation caused by passing ships. The low pass filtered time series is then subtracted from the measured time series, resulting in a time series, which is close to zero on average, but still shows the sharp peaks caused by the passing ships. This time series is then analysed. If the $NO_x$ peaks exceed a defined threshold then it is defined as an $NO_x$ plume, most likely caused from shipping. The threshold is selected to ensure that the peaks are enhancement of $NO_x$, caused
by point sources, such as ships, and not noise in the measurements. In this case, the threshold was defined as 2 ppbv. For each identified peak, the time of occurrence ($t_{peak}$), the peak width, and the height of the maximum above the background concentration are determined.

## 3.2 Ship assignment

The second step is to identify the respective source of the peak. For each peak, all ships within a 5 km radius around the measurement site up to 5 minutes before the peak maximum were investigated. For each ship, the corresponding AIS signals within the given time frame are collected and interpolated to a one second time resolution. For each AIS signal position, a trajectory is calculated to assess whether emissions caused at that specific ship position could have been transported to the measurement site by the wind. The wind speed and direction used for these trajectories are the 30 min averages of wind speed and wind direction at the measurement site. Each trajectory is calculated for the period between the time stamp of the AIS signal ($t_{AIS}$) and the time of the peak maximum ($t_{peak}$). It is then checked, whether the trajectory ends within a 50 m radius of the measurement site. If only the trajectories of a single ship end close to the measurement site in the selected time window, then this ship is assigned to be the source of the $NO_x$ peak. If no trajectory ends within the 50 m radius, the peak is considered to be caused by another source than a ship and is not analysed further. For cases, where several ships are identified as possible sources of the peak, these peaks are not analysed further, because unambiguous assignment of a single ship, as the source of the $NO_x$ emission, is not feasible. Once a ship has been identified as the source of the $NO_x$ peak, the relevant information (e.g. position, course and speed) for that particular ship passage is assigned to the peak. The first assigned ship position is the position transmitted 180 seconds before $t_{AIS}$ and the last assigned position is the position 180 seconds after $t_{peak}$. The section of these start and end points ensures that the entire $NO_x$ emission plume form a particular ship, during its passage across the measurement site, is recorded.

## 3.3 Calculation of emission rate

In the third step, the $NO_x$ emission rate for each peak assigned to a source ship is calculated. As the stations only measure the concentration of $NO_x$ at the measurement site and not at the stack of the ship, a model has to be applied to estimate the emission rate from the concentration enhancement found at the measurement site. The method, which we have chosen, is to assume that the plume of the ships can be described by a Gaussian-puff-model (Zenger, 1998):

$$C(x,y,z) = \sum_{i=1}^{N} \frac{Qdt}{\sigma_x \sigma_y \sigma_z (2\pi)^{1.5}} \cdot \exp\left(\frac{-(x - U \cdot (t - dt))^2}{2\sigma_x^2}\right) \cdot \exp\left(\frac{-y^2}{2\sigma_y^2}\right) \cdot \left[exp\left(\frac{-(z-H)^2}{2\sigma_z^2}\right) + exp\left(\frac{-(z+H)^2}{2\sigma_z^2}\right)\right] \quad (1)$$

where the concentration at a point ($C(x,y,z)$) is assumed to be a function of the emission rate ($Q$), the dispersion due to atmospheric stability ($\sigma_x$, $\sigma_y$, $\sigma_z$), the length of time of the emission ($dt$) at a certain source point (x=0, y=0), funnel height ($H$), the total transport time ($t$) and the wind speed ($U$). The wind direction is taken to be along x. The funnel height is assumed to be 5 m above the mean water level, which was always assumed to be at the surface level (z=0). The height of 5 m was chosen, because it is assumed, that the plume quickly bends down due to wind and movement of the ship. The height was always assumed to be the same, as most inland ships share a similar distinctive form, with the funnel at the back of the ship in similar heights. The model releases a puff of pollutants at the ship's position, which is then transported by the wind for an amount of time ($t - dt$) and dispersed according to the current atmospheric stability. The time ($t$) is different for each ship

position and is always the time of the last AIS signal of the ship passage ($t_{peak}$ + 180 seconds) minus the time of the respective AIS signal ($t_{AIS}$). The result is a concentration field caused by the emission of pollutants at the specific ship location for a time step $dt$. This procedure is then repeated for all ship positions. The calculated concentration fields then describe how the plume developed during the ship passage (e.g. Figure 3).

As the emission rate is unknown, the model is run with an arbitrary but constant emission rate ($Q_{model}$). The height of the plume centre is approximated to be at the height of the funnel above water level, assuming that the plume quickly bends down due to wind and movement of the ship. It is also assumed that this height is roughly the same for all ships. Dispersion parameters are chosen according to atmospheric stability, which has been determined using the wind speed at the measurement site and incoming global radiation (DWD Climate Data Center, a) during day and cloud coverage (DWD Climate Data Center, b) during night from a nearby weather station of the German Weather Service located at the Düsseldorf-Airport. To derive the emission rate, the integrated measured concentration, i.e. the area under the peak ($C_{meas}$), which has been corrected for the fluctuating background, is compared to the modelled concentration at the measurement site, i.e. the area under the modelled peak ($C_{model}$). Assuming the model sufficiently describes the ships plume, the only difference between modelled concentration and measured concentration is caused by the different emission rate. Consequently, the emission rate of the ship ($Q_{m}eas$) is estimated by the following equation:

$$Q_{meas} = \frac{C_{meas}}{C_{model}} \cdot Q_{model} \tag{2}$$

This approach assumes, that the emission rate is constant for the whole modelled time domain. An example is shown in Figure 4. In contrast to Krause et al. (2021), the whole ship passage is modelled which allows to model the transport and dispersion of the emitted trace gases as a function of time. Consequently, the peak area can be used to derive the respective emission rate of the ship, whereas in Krause et al. (2021) only the peak maximum was identified and compared with the modelled maximum to derive the emission rate. In comparison, using the peak area is a more reliable measure than the peak maximum, because it relates the total amount of pollutants arriving at the measurement site.

### 3.4 Quality control

The assumptions made in the model to estimate the $NO_x$ emission rate may not truly represent the conditions at the time of measurement. To assess the quality of the derived emission rate, Monte-Carlo-simulations are performed to assess whether a small change in one of the input parameters results in a large change of the derived concentration at the measurement site. The parameters varied are wind speed, wind direction, atmospheric stability and the position of the ship in longitude, latitude and height. Each of these parameters is changed within the uncertainty ranges given in Table 2 (an example can be found in Appendix C). For each changed parameter, the derived integrated peak concentrations are then compared to the integrated peak concentration of the reference simulation. The resulting concentrations of a set of Monte-Carlo-simulations are then summarized by the mean value ($mean_{Cj}$), the standard deviation ($\sigma_{Cj}$) and the minimum ($min_{Cj}$) and maximum value

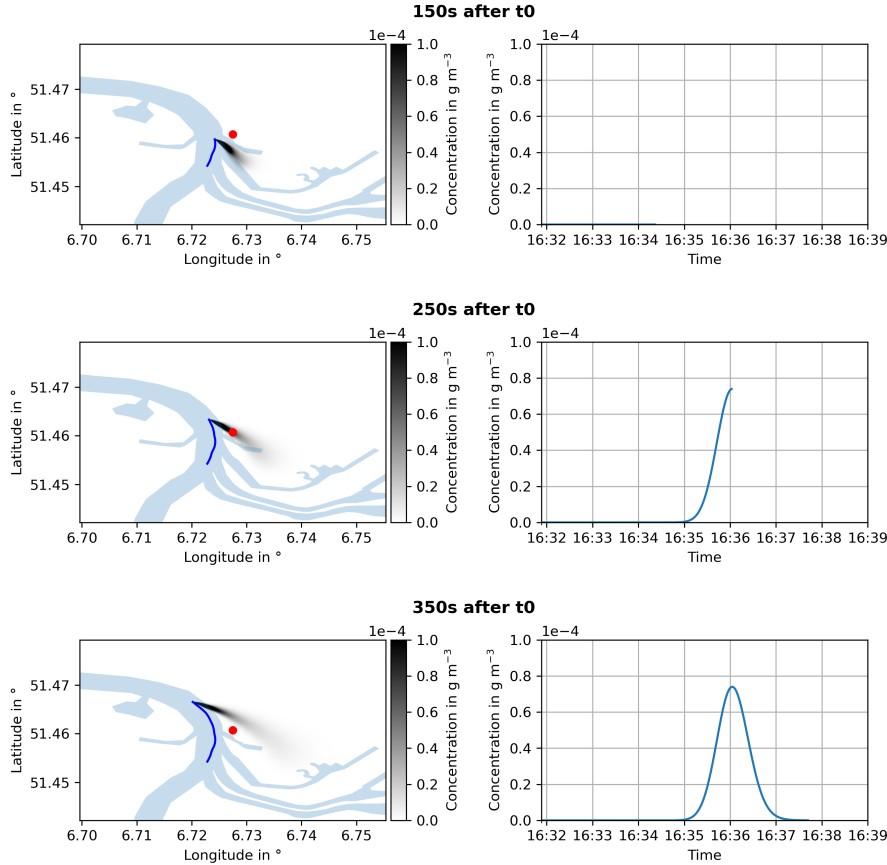

**Figure 3.** Example of a plume simulation for different time steps after the simulation start (t0). The upper, middle and lower panels show the movement of the modelled ship plume 150 s, 250 s and 350 s after the initiation of the plume. The left column shows a horizontal cross section of the modelled plume in 20 m height. The location of the measurement station is marked as a red dot. The blue line in the right column shows the modelled concentration at the location of the measurement station during the model run.

($max_{Cj}$). These values are compared to the reference simulation of the unperturbed input parameters. To be evaluated further, the following five criteria must be met by the set of Monte-Carlo-Simulations for each input parameter:

1) $mean_{Cj}$ / $C_{model}$ must be between 0.5 and 1.5, to eliminate cases with a systematic deviation caused by the uncertainty of a single input.

2) $\sigma_{Cj}$ / $C_{model}$ must be lower or equal to 1, to eliminate cases with a high variability caused by the uncertainty of a single input.

3) The difference between $min_{Cj}$ / $C_{model}$ and $max_{Cj}$ / $C_{model}$ must be smaller than 2, to eliminate cases with a large spread between minimum and maximum of the set due to the uncertainty of a single input.

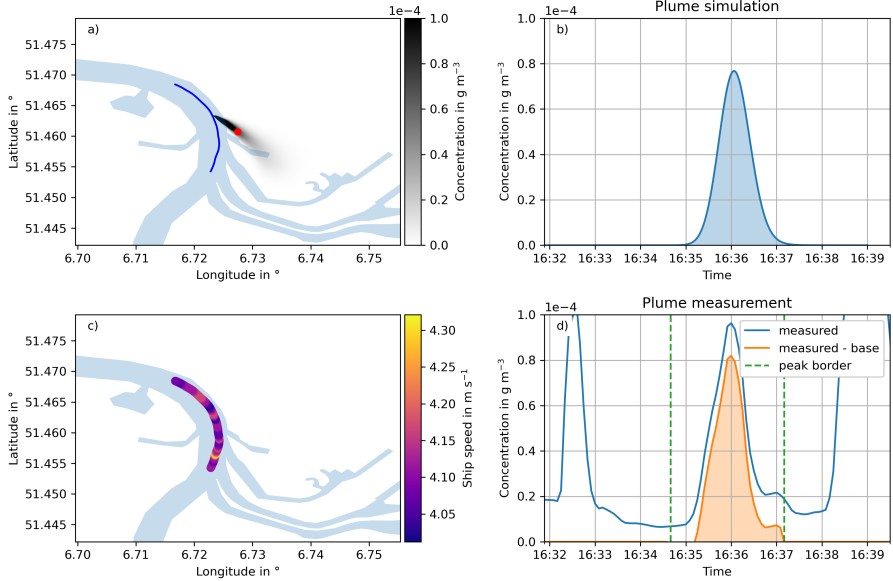

**Figure 4.** An example of a plume simulation for the 22nd August 2018 at 16:36 UTC compared with the measured plume. a) a map of the modelled plume for the time when the highest concentration has been measured. b) a plot of the simulated concentration of $NO_x$ at the measurement site as a function of time. c) a map showing the ship speed over ground for each time step. d) a plot of the measured $NO_x$ concentration as a function of time at the measurement station. The blue line represents the $NO_x$ concentration, and the orange line is the background corrected $NO_x$ concentration of the peak.

4) the absolute error of the derived emission rate must be lower than $5 \ \mathrm{g\,s^{-1}}$, which eliminates cases, where the uncertainty is on a larger order of magnitude than the emission rate.

5) the relative error of the derived emission rate must be smaller than 200 %, which eliminates cases, where the uncertainty is much larger than the emission rate.

### 3.5 Uncertainty of the $NO_x$ emission rates

In this section we investigate the uncertainty of the measurement. This is considered to be the standard error of the emission rate, i.e. one standard deviation of the distribution of emission rates. The uncertainty of the derived emission rate is given by:

$$\sigma_Q = \sqrt{\left(\frac{\partial Q_{meas}}{\partial C_{meas}} \cdot \sigma_{C\,meas}\right)^2 + \left(\frac{\partial Q_{meas}}{\partial C_{model}} \cdot \sigma_{C\,model}\right)^2} \tag{3}$$

where $\sigma_{Cmeas}$ is the uncertainty of the measured integrated peak trace gas concentration and $\sigma_{Cmodel}$ is the uncertainty of the modelled integrated peak trace gas concentration. The uncertainty of the model is defined as:

**Table 2.** Uncertainties of the input parameters used in the Monte-Carlo-Simulations.

| Abbreviation | Name | Calculation of value |
|---|---|---|
| $\sigma_{lon}$ | source position longitude | Uncertainty of the AIS signal, 10 m |
| $\sigma_{lat}$ | source position latitude | Uncertainty of the AIS signal, 10 m |
| $\sigma_H$ | plume height | $\sqrt{\sigma_{fh}^2 + \sigma_{wl}^2}$ |
| $\sigma_{fh}$ | funnel height | estimated: 5 m |
| $\sigma_{wl}$ | water level | mean high water level - mean low water level |
| $\sigma_U$ | wind speed | standard deviation of the wind speed |
| $\sigma_\theta$ | wind direction | estimated: 10 $^\circ$ |
| $\sigma_{stability}$ | stability | atmospheric dispersion parameters of class with lower stability and higher stability than the assigned class |
| $\sigma_{c_{meas}}$ | uncertainty of the measured peak area | $\sqrt{std(peak)^2 \cdot n}$, where n is the number of nodes used to calculate the peak area |

$$\sigma_{C\,model} = \sqrt{\sigma_{C\,U}^2 + \sigma_{C\,\theta}^2 + \sigma_{C\,stability}^2 + \sigma_{C\,lon}^2 + \sigma_{C\,lat}^2 + \sigma_{C\,H}^2} \qquad (4)$$

where each $\sigma_{Cj}$ is the standard deviation of the modelled trace gas concentrations of the Monte-Carlo-simulations with
205 respect to changes of an individual input parameter ($j$). In the Monte-Carlo-simulations, each parameter is varied individually, i.e. independently. The consequences of the changes of more than one parameter at a time are assumed to be negligible. The largest sources of uncertainty of the derived emission rate are the wind speed, wind direction and stability. Wind speed and wind direction influence the shape and the time of appearance of the modelled peak. The area of the peak changes as a function of wind speed, lower wind speeds lead to a larger peak, while higher wind speeds lead to a smaller peak. Also the peaks shift
in time. With lower wind speeds than assumed, the modelled plume arrives at the measurement site later than expected, while with higher than assumed wind speeds, it arrives too early. The wind direction has similar effects and also changes the peak area and the time of arrival of the peak maximum. Stability however only changes the modelled peak area, more unstable conditions lead to smaller modelled peaks, as the plume can also grow vertically and the pollutants are dispersed over a larger volume. In contrast more stable conditions leading to larger modelled peaks, as the vertical dispersion is hindered. The source
position does not play such an important role and neither the changes in latitude, longitude or height show significant changes of the modelled peak within the considered uncertainties. Also the resulting uncertainty within the measured peak area is small compared to the uncertainty caused by the wind speed, wind direction and stability. An example of the Monte-Carlo-simulations and the respective influence on the modelled peaks can be found in the Appendix in Figure C1.

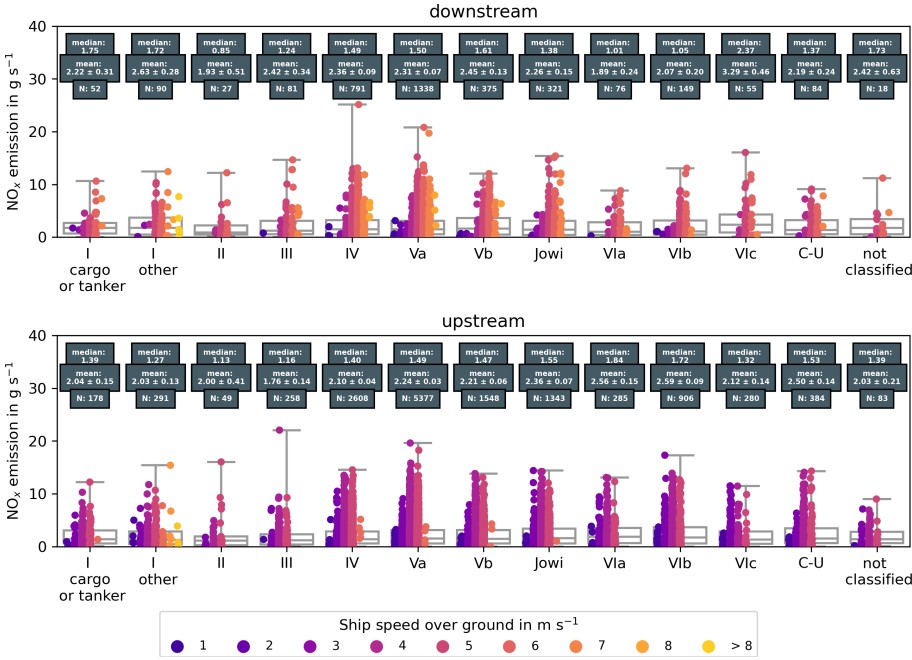

**Figure 5.** NO$_x$ emission rates for all ship classes, derived from measurements at DURH. Single measurements are colour-coded to the respective mean ship speed during the measurement.

## 4 Results

At DURH, more than 291000 ship passages were identified in the AIS signals. For 32900 ship passages peaks have been identified and could be assigned to specific source ships. For 23500 of those peaks it was possible to determine the NO$_x$ emission rate, which fulfil the criteria of the quality control described previously. At NERH, 5500 peaks have been identified and the respective emission rates have been derived, in 3200 cases those derived NO$_x$ emission rates fulfil the quality criteria. The number of identified ship plumes is mainly limited by the wind, as the wind is needed to transport the emitted pollutants towards the measurement site. An additional limitation is the traffic density as in situations of high traffic, an unambiguous identification of a ship plume is often not possible.

### 4.1 DURH

The derived emission rates were then summarized in the context of the respective CEMT ship class (Table 3), the direction of travel (upstream or downstream) and their speed over ground (e.g. Figure 5). The majority of ships belong to ship classes IV, Va, Vb and Jowi, which together account for approximately 80 percent of the total ship traffic (Figure 6). Between 2017 and 2021, there were approximately 256 ship passages each day. As can be seen in Figure 7, the majority ships travelling upstream have a speed over ground of about 3 m s$^{-1}$, while the majority of ships travelling downstream have speeds over ground of about 5 m s$^{-1}$.

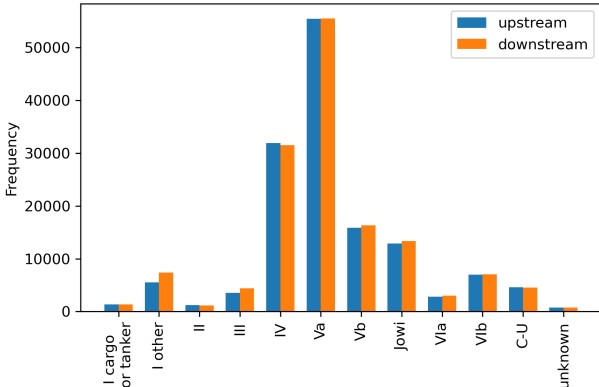

**Figure 6.** Ship traffic and fleet composition at DURH between November 2017 and December 2021. In total 291635 ship passages have been identified.

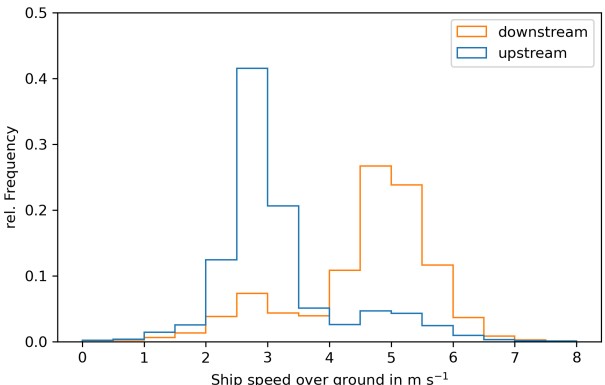

**Figure 7.** Ship speed over ground for all ship passages identified at DURH as a function of direction of travel.

For the most common ship classes, this enables the $NO_x$ emission rates of the respective class under real driving conditions to be characterized. For less common ship classes, there are fewer observations, which leads to a higher uncertainty of the summarized $NO_x$ emission rates for these classes. In addition, there might not be enough data to differentiate sufficiently between direction of travel or different speeds. The speed over ground is correlated with the emission rates, higher speeds leading to higher emissions as expected (e.g. Figure 8).

Furthermore, the direction of travel is important when investigating the emissions for a given speed. Ships that travel upstream have to overcome the river current and therefore need more power to achieve the same speeds over the ground compared to ships travelling downstream. With the same speed in water, the engine operating conditions should be similar and independent of direction of travel. Consequently, we assume that the $NO_x$ emission rates are similar. A direct comparison for ship classes IV, Va, Vb and Jowi shows, that ships travelling upstream with a speed of about $3 \, \mathrm{m \, s^{-1}}$ and ships travelling down-

stream with a speed of $5\,\mathrm{m\,s^{-1}}$ have similar $NO_x$ emission rates in their respective size class (shown in Table 5), which suggests similar operating conditions.

Unfortunately, at the DURH station most of the identified ships are vessels which are travelling upstream. Out of the 23500 quality checked emission rates, approximately 13500 are emitted by ships travelling upstream, approximately 3400 are from ships travelling downstream and 6500 changed their direction within the modelled time frame, i.e. to enter or leave DURH or a further connection to a channel. The main wind direction at DURH is south-west which is parallel to the river, and ship plumes are therefore transported along the river. Unambiguous assignment is only possible if there is just a single ship plume that can reach the measurement station. Ships travelling upstream need a longer time to pass through the area, as they are slower than ships travelling downstream. Therefore, in cases of high traffic density, the longer time window of the slower upstream travelling ships increases the chances of an unambiguous identification and results in a larger number of observed ship plumes for that particular direction.

The $NO_x$ emission rates in the context of size (or ship class) are more difficult to summarize (see Figure 5). Generally, larger ships show larger $NO_x$ emission rates than smaller ships. Larger ships usually have more powerful engines to provide the power needed to move and manoeuvre the ship. More powerful and larger engines consume more fuel and therefore have higher emission rates than smaller engines. At the same time, the larger ships are usually newer and their emissions are regulated, while older ships are subject to grandfathering, which means their engines do not have to comply with new regulations. Only if the engine of an older ship is exchanged, are the new regulations applicable. Due to the long service life of inland ships, a lot of the smaller ships do not fall under the regulations and therefore still have high emissions.

### 4.1.1 Comparison with on-board emission measurements

In order to validate the emission rates within the CLINSH project, a comparison has been carried out between the values derived here from on-shore observations of the CLINSH fleet and the respective on-board measurements. CLINSH ships have been identified using the AIS signal as described in section 3. For the $NO_x$ plumes from shipping, which passed the quality control criteria, the CLINSH data was searched to identify whether on-board data are available for the same time interval. For the case of a match, on-board data have been averaged for the period, in which the plume detected by the on-shore observation system was released by the ship. As the uncertainty of the Gaussian-puff-model is quite high, data one minute before and after the release time were included in the plume average to take this into account. The 16 different CLINSH ships were observed nearly 200 times with both on-board and on-shore measurement systems. Table 4 and Figure 9 give a summary of the results obtained.

For almost half of the ships, the agreement between on-board and on-shore observations is good and well within the error bars. However, it turns out that for some ships (e.g. ship M), on-shore values are systematically higher than the on-board data for the same time. One possible explanation is that some ships use more than one main engine for navigation, but the on-board measurement systems usually only capture the emissions of one of the engines and not the total amount emitted at the stack. The total emission rate for all main engines is assumed to be the number of engines multiplied with the measured on-board emission rates. In addition some vessels also use auxiliary engines to power generators or bow thrusters, which also add to

**Table 3.** Modified ship classification scheme based on CEMT (European Conference of Ministers of Transport, 1992) classes. Ships are categorized by their respective length and width, e.g. a ship longer than 86 m but shorter than 111 m and width between 10 and 12 m is classified as class Va. Additionally coupled units are identified via their Electronic Reporting International (ERI) code which is also transmitted in the AIS signals.

| Class | maximum length | maximum width | cargo capacity |
|---|---|---|---|
| | m | m | tons |
| I | 39 | 6 | 350 |
| II | 56 | 7 | 655 |
| III | 68 | 9 | 1000 |
| IV | 86 | 10 | 1350 |
| Va | 111 | 12 | 2750 |
| Vb | 136 | 12 | 4000 |
| Jowi | 136 | 18 | 5300 |
| VIa | 173 | 12 | 5500 |
| VIb | 194 | 23 | 11000 |
| VIc | 194 | 35 | 16500 |
| Coupled unit (C-U) | motor freighter pushing one barge identified via ERI identifier | | |
| unknown | ships without information about width and / or length | | |

the total emissions of the ship and can seen by the on-shore measurements but not the on-board measurements. Taking into account all ships and all simultaneous observations, the ratio between on-shore and on-board is about $1.3 \pm 0.1$ (see Figure 9).
Additionally, ship M is equipped with a SCRT (selective catalytic reduction) system to reduce the $NO_x$ emissions, which did not always operate.

### 4.1.2   Comparison with other studies

The emission behaviour of vessels is usually described and evaluated by emission factors. These emission factors are relative measures, e.g. the amount of emitted $NO_x$ is expressed per amount of burnt fuel or per amount of power generated by the
engine. The absolute emission rate of $NO_x$ has to be calculated from the emission factors and additional information about the fuel consumption. For comparison with the emission factors derived in other studies, two fuel consumption scenarios are considered. In the first scenario, a fuel consumption of 108 $kg\,h^{-1}$ is assumed, which describes the fuel consumption of a ship with 3200 tons cargo capacity travelling downstream. The second scenario uses a fuel consumption of 162 $kg\,h^{-1}$, which describes the fuel consumption of a ship with 3200 tons cargo capacity travelling upstream against the current. Both scenarios
are based on the specific fuel consumptions in $kg\,km^{-1}$, which are 6 $kg\,km^{-1}$ for ships travelling downstream and 15 $kg\,km^{-1}$ for ships travelling upstream (Allekotte et al., 2020). The specific fuel consumptions have been converted to $kg\,h^{-1}$ using the average speed over ground for ships travelling upstream and downstream, which are 3 and 5 $m\,s^{-1}$, respectively.

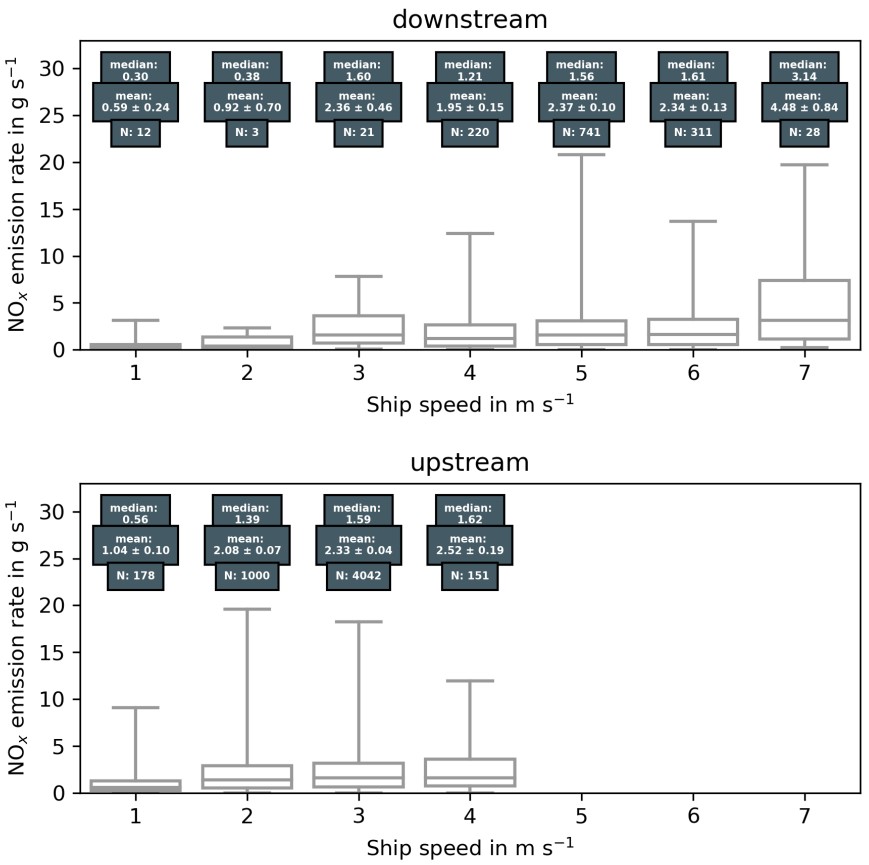

**Figure 8.** $NO_x$ emission rates for ship class Va and their dependence on the direction of travel and ship speed over ground, derived from data measured at DURH.

Table 5 shows the comparison of literature values applied to these two scenarios with the emission rates derived in this study. The lower fuel consumption scenario shows absolute $NO_x$ emission rates between $1.17 \, \mathrm{g \, s^{-1}}$ to $1.71 \, \mathrm{g \, s^{-1}}$. The higher fuel consumption scenario shows emission rates from $1.75 \, \mathrm{g \, s^{-1}}$ to $2.57 \, \mathrm{g \, s^{-1}}$. In comparison, the mean $NO_x$ emission rates derived in DURH for ships that travel downstream with the most common speed of $5 \, \mathrm{m \, s^{-1}}$ are in the range of $2.36 \, \mathrm{g \, s^{-1}}$ to $2.53 \, \mathrm{g \, s^{-1}}$. For ships travelling upstream with the most common speed over ground of $3 \, \mathrm{m \, s^{-1}}$ the $NO_x$ emission rates are 2.17 to $2.36 \, \mathrm{g \, s^{-1}}$. Generally, the mean $NO_x$ emission rates fit into the range given by the emission factors of other studies, but are at the upper limit of the given range. At lower speeds, the mean emission rates are also lower. At the most common speeds over ground in a given direction, the emission rates for ships travelling upstream and ships travelling downstream are similar. In general, the scenario with high fuel consumption seems to better reflect the derived emission rates. This indicates similar fuel consumption and engine operation scenarios for ships travelling downstream and ships travelling upstream. Assuming a

**Table 4.** Comparison of $NO_x$ emission rates derived from on-shore measurements and on-board measurements for different ships partici-pating in the CLINSH project. Number of engines only includes main engines used for navigation, and on-board measurements were only carried out on one of them. The number of engines used on ship G is not known, but assumed to be one.

| Ship | class | No. of engines | on-shore mean $\mathrm{g\,s^{-1}}$ | on-shore median $\mathrm{g\,s^{-1}}$ | on-shore std $\mathrm{g\,s^{-1}}$ | on-board mean $\mathrm{g\,s^{-1}}$ | on-board std $\mathrm{g\,s^{-1}}$ | n |
|------|-------|----------------|--------------|----------------|-------------|---------------|--------------|---|
| A | III | 1 | 0.84 | 0.84 | 0.27 | 1.23 | 0.34 | 2 |
| B | IV | 1 | 0.94 | 0.40 | 0.92 | 0.81 | 0.32 | 6 |
| C | IV | 1 | 2.20 | 1.66 | 1.29 | 1.34 | 0.51 | 6 |
| D | Va | 1 | 2.12 | 1.75 | 0.63 | 0.73 | 0.41 | 3 |
| E | Va | 1 | 0.56 | 0.56 | - | 0.42 | 0.14 | 1 |
| F | Va | 1 | 2.40 | 1.55 | 2.33 | 2.17 | 0.67 | 45 |
| G | Va | ? | 1.89 | 1.77 | 0.71 | 1.53 | 0.32 | 5 |
| H | Va | 1 | 3.65 | 3.85 | 2.44 | 2.47 | 1.23 | 4 |
| I | Jowi | 1 | 1.63 | 1.77 | 0.88 | 1.13 | 0.32 | 4 |
| J | Jowi | 1 | 2.05 | 0.30 | 3.86 | 0.71 | 0.41 | 13 |
| K | Jowi | 1 | 1.58 | 1.30 | 1.10 | 0.92 | 0.43 | 14 |
| L | C-U | 1 | 1.43 | 0.74 | 1.49 | 0.35 | 0.16 | 7 |
| M* ** | III | 2 | 2.15 | 1.70 | 2.24 | 0.65 (1.30) | 0.43 (0.86) | 13 |
| N* | Va | 3 | 1.73 | 0.98 | 1.81 | 0.61 (1.83) | 0.32 (0.96) | 9 |
| O* | Jowi | 2 | 1.56 | 0.75 | 2.08 | 0.72 (1.44) | 0.39 (0.78) | 25 |
| P* | VIb | 2 | 1.44 | 0.66 | 1.37 | 0.83 (1.66) | 0.42 (0.84) | 17 |

* Ships M, N, O and P are equipped with more than one main engine used for navigation. It is assumed that the $NO_x$ emission rates for all engines are the same. The total emission rate for all main engines is therefore assumed to be the number of engines multiplied with the measured on-board emission rates, shown in brackets.

** Ship M is equipped with a selective catalytic reduction (SCR) system to reduce the $NO_x$ emissions, which was not always operating.

water velocity of $1\ \mathrm{m\,s^{-1}}$, the average speed in water would be similar for both directions, which would also indicate similar engine operation conditions and therefore similar emission rates.

### 4.1.3 Comparison to current $NO_x$ regulations

Table 6 shows the regulations that are in place for ships built or which had their engine replaced in the specified years. The regulations are defined in $\mathrm{g\,kWh^{-1}}$ and have been converted to $\mathrm{g\,kg^{-1}}$ using a specific fuel consumption of 230 $\mathrm{g\,kWh^{-1}}$ (De Vlieger et al., 2004). The specific fuel consumption of ship engines can vary between different engines and also depends on the engine load, lower engine load generally increasing the specific fuel consumption (van Mensch et al., 2018). Measurements presented by van Mensch et al. (2018) show that the specific fuel consumption for different engines can reach about 290 $\mathrm{g\,kWh^{-1}}$ for engine loads below 20 % and between 200 and 230 $\mathrm{g\,kWh^{-1}}$ for engine loads higher than 20 %. Additionally, the specific fuel consumption also depends on the age of the engine, newer engines generally having a lower specific fuel consumption (De Vlieger et al., 2004). To interpret the derived $NO_x$ emission rates in the context of these regulations, the

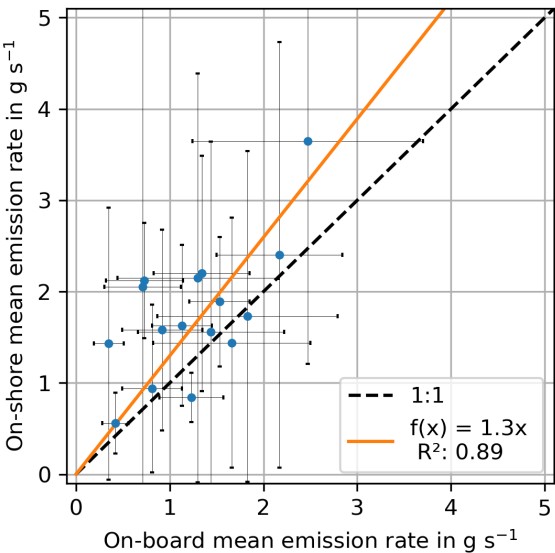

**Figure 9.** Scatter plot of on-board and on-shore emission rates. Each dot represents the mean value for one ship, errorbars indicate respective standard deviations. For ships with more than one main engine, the number of engines has been taking into account for the on-board emission rates. See also Table 4.

limits given in the regulations were converted to $\mathrm{g\,s^{-1}}$ using the $162\ \mathrm{kg\,h^{-1}}$ fuel consumption scenario. These values then can be interpreted as an upper limit for the $NO_x$ emission rates for cases of high fuel consumption. Figure 10 shows the $NO_x$ emission rates derived from the on-shore measurements at DURH for the most common ship classes (VI, Va, Vb and Jowi) as a function of their respective speed over ground. For all ship classes the mean $NO_x$ emission rates for speeds higher than 2 $\mathrm{m\,s^{-1}}$ exceed even the least strict regulation CCNR I of $9.2\ \mathrm{g\,kWh^{-1}}$. For speeds over ground lower than 3 $\mathrm{m\,s^{-1}}$ the mean $NO_x$ emission rates are within the CCNR I limit, but in these cases, the assumed high fuel consumption scenario usually does not apply. When looking at the individual ship passages for the classes IV, Va, Vb and Jowi, approximately 50 % of the derived $NO_x$ emission rates plus their respective uncertainty $(Q_{meas} + \sigma_Q)$ are below the CCNR I upper limit, approximately 40 % are below CCNR II and 16 % are below EU RL2016/1629. These results indicate that a large number of old ships with unregulated engines are still in operation.

Kurtenbach et al. (2016) reported emission factors of 20 to 161 $\mathrm{g\,kg^{-1}}$ with an average of $52 \pm 3\ \mathrm{g\,kg^{-1}}$, while Kattner (2019) derived a mean emission factor of $41 \pm 28\ \mathrm{g\,kg^{-1}}$. In both studies the mean emission factor is above the limits given by the regulations, but also here individual ships already comply with them.

In addition, it has to be kept in mind that the water level, hull form and propeller configuration can have a significant influence on the power required to navigate a ship, and therefore on the amount of emitted pollutants (Friedhoff et al., 2018). The mean $NO_x$ emission rates presented here are the result of the evaluation of several years and thousands of different ships.

**Table 5.** Comparison of the derived $NO_x$ emission rates (ER) in $g\,s^{-1}$ with the emission factors (EF) in $kg\,h^{-1}$ derived from other studies. To calculate the emission rate from the emission factors, two fuel consumption scenarios are evaluated. Both scenarios are based on specific fuel consumption values for ships with a cargo capacity of 3200 tons (approximately class Va and Vb). First a fuel consumption of 108 $kg\,h^{-1}$ is assumed for ships that travel downstream, second a fuel consumption of 162 $kg\,h^{-1}$ is assumed for ships travelling upstream.

| Study | $NO_x$ EF in $g\,kg^{-1}$ | $NO_x$ ER in $g\,s^{-1}$ | $NO_x$ ER in $g\,s^{-1}$ |
|---|---|---|---|
| Fuel consumption | | 108 $kg\,h^{-1}$ | 162 $kg\,h^{-1}$ |
| Trozzi and Vaccaro (1998) | 51 | 1.53 | 2.30 |
| Kesgin and Vardar (2001) | 57 | 1.71 | 2.57 |
| Klimont (2002) | 51 | 1.53 | 2.30 |
| Rohács and Simongáti (2007) | 47 | 1.41 | 2.12 |
| Schweighofer, J. and Blaauw, H. (2009) | 39 | 1.17 | 1.75 |
| van der Gon and Hulskotte (2010) | 45 | 1.35 | 2.03 |
| Diesch et al. (2013) | 53 | 1.59 | 2.39 |
| Umweltbundesamt (2013) | 49 | 1.47 | 2.21 |
| Kurtenbach et al. (2016) | 54 | 1.62 | 2.43 |
| Kattner (2019) | 41 | 1.23 | 1.85 |
| This study (DURH) | | downstream | upstream |
| Speed over ground | | 5 $m\,s^{-1}$ | 3 $m\,s^{-1}$ |
| IV | - | $2.36 \pm 0.13$ | $2.17 \pm 0.05$ |
| Va | - | $2.37 \pm 0.10$ | $2.33 \pm 0.04$ |
| Vb | - | $2.53 \pm 0.17$ | $2.35 \pm 0.07$ |
| Jowi | - | $2.26 \pm 0.19$ | $2.36 \pm 0.08$ |

**Table 6.** Overview of $NO_x$ emission limits, according to CCNR (EUD, 1998; CCNR, 2020) and EU regulations (EUR, 2016), in both cases given in units of $g\,kWh^{-1}$. For comparison these have been converted to $g\,kg^{-1}$ using a specific fuel consumption for inland ships of 230 $g\,kWh^{-1}$ (De Vlieger et al., 2004) and eventually to $g\,s^{-1}$ using the 162 $kg\,h^{-1}$ fuel consumption scenario.

| Regulation | in effect since | Engine power in kW | $NO_x$ EF in $g\,kWh^{-1}$ | $NO_x$ EF in $g\,kg^{-1}$ | $NO_x$ ER in $g\,s^{-1}$ |
|---|---|---|---|---|---|
| CCNR I | 2002 | P > 130 | 9.2 | 39.9 | 1.80 |
| CCNR II | 2007 | P > 130 | 6.0 | 26.1 | 1.17 |
| EU RL2016/1629 | 2019 | 130 < P < 300 | 2.1 | 9.1 | 0.41 |
| EU RL2016/1629 | 2019 | P > 300 | 1.8 | 7.8 | 0.35 |

It is therefore expected that the mean values are representative for the average ship emissions on the Rhine at the DURH measurement site.

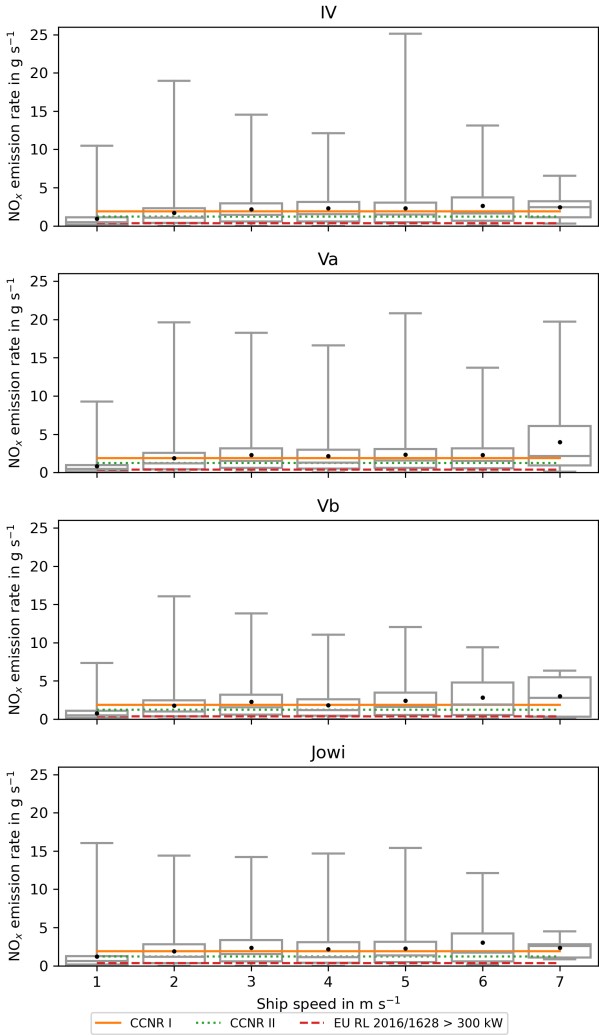

**Figure 10.** Boxplots of $NO_x$ emission rates for ship classes IV, Va, Vb and Jowi as a function of ship speed over ground, derived from data measured at DURH. Mean value is shown as a black dot, median value as grey line. The limits given by the CCNR I, CCNR II and EU RL2016/1628 regulations were converted from $g\,kWh^{-1}$ to $g\,s^{-1}$ and are shown as lines (see Table 6 for more details).

In addition to regulation of new ships and engines, additional technical measures, such as exhaust gas after-treatment can be used to reduce the emissions caused by ship traffic. The capabilities of exhaust gas after-treatment systems has already been discussed in previous studies (e.g., Schweighofer, J. and Blaauw, H., 2009; Kleinebrahm and Bourbon, 2013; Pirjola et al., 2014; Brandt and Busch, 2017; Busch et al., 2020).

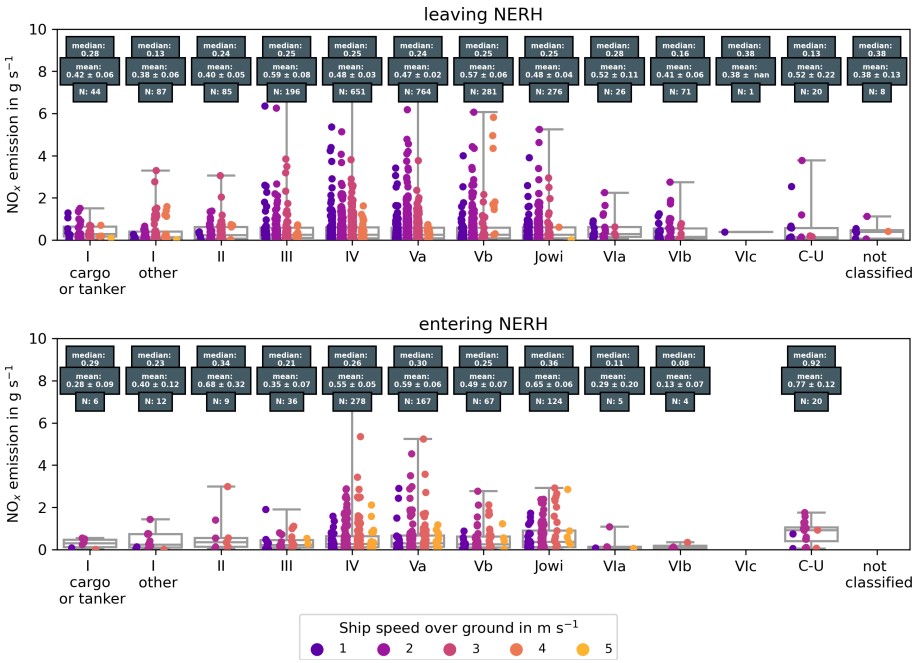

**Figure 11.** $NO_x$ emission rates for all ship classes, derived from measurements at NERH. Single measurements are colour-coded to the respective mean ship speed during the measurement.

## 4.2 NERH

As the NERH measurement site is located directly within the harbour area, the ships here are generally slower and show lower $NO_x$ emission rates compared to the DURH measurement site (e.g. Figure 11). As there is no strong river current, ships have been classified into ships leaving the harbour and ships entering the harbour, instead of travelling upstream or downstream. Generally, ships show similar emission rates independent of them leaving or entering the harbour area (e.g. Figure 12).

## 4.3 Ideal measurement location

The improved algorithm presented here, has several advantages over the method described in Krause et al. (2021), where a Gaussian-plume-model was used to derive $NO_x$ and $SO_2$ emission rates from Long-Path DOAS measurements. An in-situ station is easier to model than a remote sensing site, because the concentration is only measured at the location of the station and does not represent the integrated column of an absorber along a light path. The equipment used in this study can be found in standardized air quality measurement stations, facilitating the use of existing stations for ship emission estimates. Only the additional AIS receiver is needed to provide information about the passing vessels. This means that $NO_x$ emission rates can be derived from existing stations with little additional costs. In addition, in-situ measurement stations are able to measure NO and $NO_2$ simultaneously, so that $NO_x$ can be measured directly and has not to be inferred from $NO_2$ observations as in Krause et al. (2021).

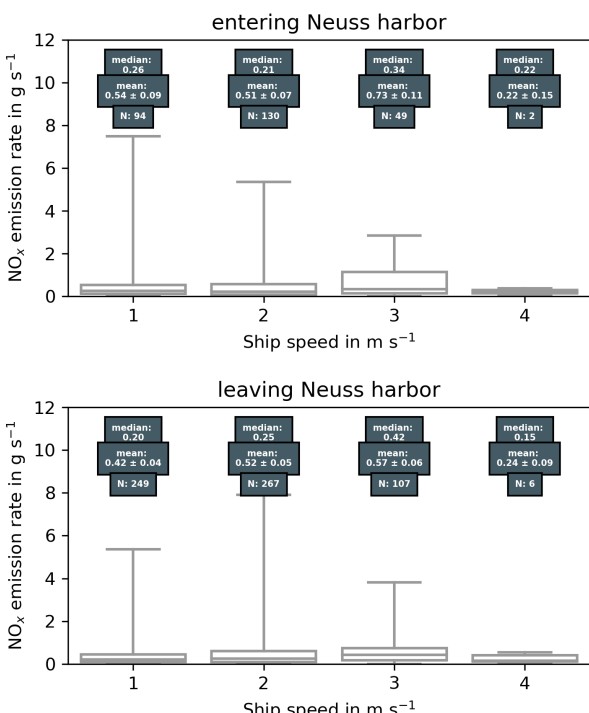

**Figure 12.** $NO_x$ emission rates for ship class IV and their dependence on the direction of travel and ship speed over ground, derived from data measured at NERH.

The measurement stations in DURH and NERH were both suitable locations to derive emission rates from passing vessels under real ship driving conditions. However, their locations are not ideal and increase the difficulty when applying the algorithm to the measurement data. At the time of the installation of the measurement sites, the derivation of on-shore emission rates was not the focus of the CLINSH project. Consequently, we consider that optimisation of the position of the measurement can

improve the derivation of the emission rates and lower its uncertainty.

Ideally, a measurement station would be located at a section of a river where there are no confluences. This helps in analysing the derived emission rates, as it is easier to distinguish between ships travelling upstream and downstream. Also it removes possible special manoeuvres carried out by the ships trying to enter or leave a confluence. Further, the measurement station should be located at a straight river section, preferably with the main wind direction orthogonal to the river. This decreases the

chances of overlapping plumes and therefore increases the chances to identify the source ship. Locations where the wind blows along the river should be avoided, because the plumes of several ships can be mixed and the identification of the source ships can become impossible, especially when there is high traffic. Locations with point sources of $NO_x$ upwind of the measurement site should also be avoided. These point sources could cause additional peaks, mix with the ship plumes and alter their respective peaks in the measured time series or simply lead to a highly variable background concentration which might be hard to correct.

The terrain around the measurement site should be flat and even, so that the surface roughness can be characterized easily. In summary, a simple geometry of the surroundings and a low number of obstacles (i.e. trees, buildings) is beneficial when using the Gaussian-puff-model. In addition, usage of measurements of the current water level would be beneficial because the uncertainty in the height of the emission could be reduced. Incoming solar radiation and cloud cover should ideally be measured at the measurement site, to reduce the uncertainty regarding these parameters.

These suggestions about making emission measurements are not required to the derive the emission rates, as has been shown in this study, but using them will improve the accuracy of future measurements.

## 5 Conclusions

As part of this study, two standardized in-situ measurement stations have been set up to measure ship emissions on the river Rhine. The first was set up on the river shore in Duisburg (DURH) to measure the emissions directly at the Rhine, while the
375 second one was installed in the harbour area of Neuss (NERH). The measurement stations were established in the period of September to October 2017. The station at DURH is still active while that at NERH made its planned measurements and was dismantled at the end of 2019. For both stations it was possible to identify peaks in the measured $NO_x$ time series and find the corresponding source ships. A new method to derive absolute emission rates (in $g\,s^{-1}$) from these peaks was developed and successfully applied to the data. Within the algorithm, each individual ship passage is modelled by a Gaussian-puff-model and
380 the modelled concentration at the measurement site is compared to the measured concentration to calculate the emission rate. The modelled concentrations are quality controlled for non-physical results, which can occur when the uncertainty of the input parameters used in the Gaussian-puff-model is too high. At DURH approximately 32900 peaks have been identified and could be attributed to a source ship and in approximately 23500 cases, quality controlled emission rates were derived. At NERH, approx. 5500 peaks have been identified and approx. 3200 emission rates were derived. These emission rates were analysed
in context of ship class (size), speed over ground and direction of travel (upstream and downstream). Generally, the emission rates increase with ship size and ship speed, also the emission rates of ships travelling upstream are higher than those of ships travelling downstream, but have the same speed over ground.

The derived emission rates in this study have been compared to emissions rates measured on-board of ships that participated in the project, and generally good agreement between both methods was found. Discrepancies can be explained by the different
quantities that are measured. The on-shore measurements represent the sum of all $NO_x$ emissions of the ship, including all auxiliary engines, while the on-board measurements are only carried out on the main engine. For ships which use more than one engine for navigation, the on-board measurements were only realised for one engine and not for all of them. Therefore, the number of engines had to be considered for the comparison of on-shore and on-board measurements. The emission rates have been compared to emission factors (in $g\,kg^{-1}$) from other studies, under the assumption of two fuel consumption scenarios,
and agree quite well considering the uncertainties.

The mean emission rates for the most common ship classes (IV, Va, Vb and Jowi) at speeds higher than $2\,\mathrm{m\,s^{-1}}$ exceed even the least strict regulations of CCNR I of $9.2\,\mathrm{g\,kWh^{-1}}$. Looking at individual ship passages for these four classes, approximately 50 % comply with CCNR I, 40 % comply with CCNR II and 16 % comply with EU RL2016/1629.

The algorithm mostly relies on input parameters that are routinely measured by standardized air quality stations, only additional information about the passing ships is needed and can be provided by AIS receivers. In contrast to emission factors, the derived emission rates can be directly used in the conjunction with traffic statistics to model the total emissions caused by ship traffic in the area. This enables possible uncertainties, caused by the assumptions made to convert relative emission factors to absolute emission rates during the modelling process, to be circumvented. In addition, the emission rates include the emission of all engines on-board the ships and not only of the main engine for each passing vessel.

Generally, the derived emission rates should be representative for the lower Rhine area, with similar streaming conditions as they are encountered at the DURH measurement site. At the same time, evaluation of the AIS signals derived from DURH and NERH show, that ships tend to adapt their speed to streaming conditions encountered at each measurement site, which could also influence their emission rates.

The emission rates collected in 2017-2021 have already been applied by LANUV for the port areas of DURH and NERH within the framework of CLINSH to calculate shipping-related emissions. It is planned to use this procedure for the future update of the inland waterway vessel emission register of the state of North Rhine-Westphalia for the determination of shipping emissions. The continuously measuring station at DURH will remain in operation in the coming years and will be evaluated using the described algorithm.

*Code and data availability.* The data and code used in this study is available directly from the authors upon request. The derived $NO_x$ emission rates can be found in the supplements of this paper.

# Appendix A: $NO_x$ emission rates DURH

**Table A1.** Summary of data shown in Figure 5, $NO_x$ emission rates for ships travelling upstream derived from measurements at DURH.

| Class | Mean in $g\,s^{-1}$ | Median in $g\,s^{-1}$ | Min in $g\,s^{-1}$ | Max in $g\,s^{-1}$ | Std in $g\,s^{-1}$ |
|---|---|---|---|---|---|
| Coupled unit (C-U) | 2.50 | 1.53 | 0.001 | 14.31 | 2.75 |
| I cargo or tanker | 2.04 | 1.39 | 0.009 | 12.21 | 2.03 |
| I other | 2.03 | 1.27 | 0.006 | 15.4 | 2.27 |
| II | 2.0 | 1.13 | 0.007 | 16.04 | 2.9 |
| III | 1.76 | 1.16 | 0.003 | 22.07 | 2.2 |
| IV | 2.1 | 1.4 | 0.001 | 14.54 | 2.2 |
| Jowi | 2.36 | 1.55 | 0.001 | 14.4 | 2.44 |
| VIa | 2.56 | 1.84 | 0.001 | 13.09 | 2.59 |
| VIb | 2.59 | 1.72 | 0.001 | 17.31 | 2.7 |
| VIc | 2.12 | 1.32 | 0.004 | 11.5 | 2.35 |
| Va | 2.24 | 1.49 | 0.002 | 19.62 | 2.32 |
| Vb | 2.21 | 1.47 | 0.002 | 13.84 | 2.24 |
| not classified | 2.03 | 1.39 | 0.031 | 9.0 | 1.9 |

**Table A2.** Summary of data shown in Figure 5, $NO_x$ emission rates for ships travelling downstream derived from measurements at DURH.

| Class | Mean in $g\,s^{-1}$ | Median in $g\,s^{-1}$ | Min in $g\,s^{-1}$ | Max in $g\,s^{-1}$ | Std in $g\,s^{-1}$ |
|---|---|---|---|---|---|
| Coupled unit (C-U) | 2.19 | 1.37 | 0.054 | 9.12 | 2.18 |
| I cargo or tanker | 2.22 | 1.75 | 0.020 | 10.61 | 2.23 |
| I other | 2.63 | 1.72 | 0.007 | 12.41 | 2.70 |
| II | 1.93 | 0.85 | 0.085 | 12.2 | 2.65 |
| III | 2.42 | 1.24 | 0.004 | 14.64 | 3.04 |
| IV | 2.36 | 1.49 | 0.002 | 25.13 | 2.59 |
| Jowi | 2.26 | 1.38 | 0.0005 | 15.39 | 2.67 |
| VIa | 1.89 | 1.01 | 0.020 | 8.8 | 2.07 |
| VIb | 2.07 | 1.05 | 0.017 | 13.06 | 2.4 |
| VIc | 3.29 | 2.37 | 0.052 | 16.04 | 3.41 |
| Va | 2.31 | 1.50 | 0.003 | 20.81 | 2.52 |
| Vb | 2.45 | 1.61 | 0.001 | 12.04 | 2.49 |
| not classified | 2.42 | 1.73 | 0.062 | 11.19 | 2.69 |

## Appendix B: NO$_x$ emission rates NERH

**Table B1.** Summary of data shown in Figure 11, NO$_x$ emission rates for ships entering NERH.

| Class | Mean in g s$^{-1}$ | Median in g s$^{-1}$ | Min in g s$^{-1}$ | Max in g s$^{-1}$ | Std in g s$^{-1}$ |
|---|---|---|---|---|---|
| Coupled unit (C-U) | 0.77 | 0.92 | 0.043 | 1.75 | 0.52 |
| I cargo or tanker | 0.28 | 0.29 | 0.009 | 0.54 | 0.22 |
| I other | 0.40 | 0.23 | 0.004 | 1.43 | 0.43 |
| II | 0.68 | 0.34 | 0.032 | 2.98 | 0.96 |
| III | 0.35 | 0.21 | 0.006 | 1.90 | 0.40 |
| IV | 0.55 | 0.26 | 0.001 | 7.48 | 0.81 |
| Jowi | 0.65 | 0.36 | 0.001 | 2.91 | 0.72 |
| VIa | 0.29 | 0.11 | 0.055 | 1.07 | 0.44 |
| VIb | 0.13 | 0.08 | 0.022 | 0.34 | 0.15 |
| Va | 0.59 | 0.30 | 0.001 | 5.24 | 0.83 |
| Vb | 0.49 | 0.25 | 0.006 | 2.76 | 0.60 |

**Table B2.** Summary of data shown in Figure 11, NO$_x$ emission rates for ships leaving NERH.

| Class | Mean in g s$^{-1}$ | Median in g s$^{-1}$ | Min in g s$^{-1}$ | Max in g s$^{-1}$ | Std in g s$^{-1}$ |
|---|---|---|---|---|---|
| Coupled unit (C-U) | 0.52 | 0.13 | 0.001 | 3.77 | 0.97 |
| I cargo or tanker | 0.42 | 0.28 | 0.029 | 1.50 | 0.37 |
| I other | 0.38 | 0.13 | 0.0004 | 3.30 | 0.58 |
| II | 0.40 | 0.24 | 0.001 | 3.05 | 0.48 |
| III | 0.59 | 0.25 | 0.001 | 8.57 | 1.05 |
| IV | 0.48 | 0.25 | 0.0001 | 7.91 | 0.72 |
| Jowi | 0.48 | 0.25 | 0.003 | 5.24 | 0.70 |
| VIa | 0.52 | 0.28 | 0.061 | 2.25 | 0.54 |
| VIb | 0.41 | 0.16 | 0.007 | 2.75 | 0.52 |
| Va | 0.47 | 0.24 | 0.0004 | 7.02 | 0.69 |
| Vb | 0.57 | 0.25 | 0.001 | 6.06 | 0.92 |
| not classified | 0.38 | 0.38 | 0.035 | 1.12 | 0.36 |

## Appendix C: Monte-Carlo simulation

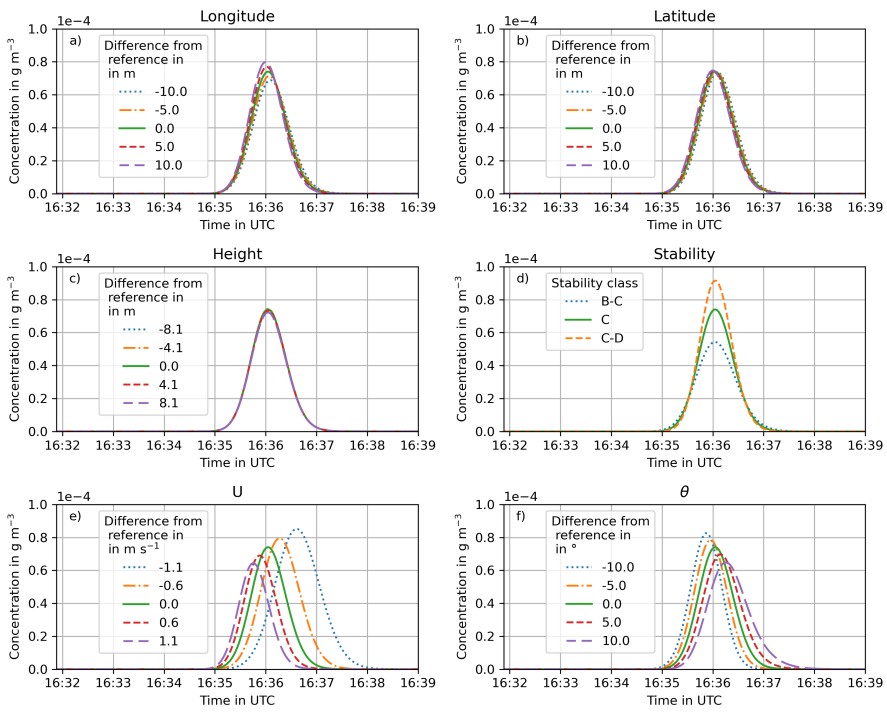

**Figure C1.** Monte-Carlo simulations for the example plume shown in Figure 4. In each plot one parameter has been changed within its respective uncertainty and the resulting model peaks are shown. The uncertainties shown in the plot legends are always expressed as deviation from the reference simulation, e.g. in plot a) -10 m means that the ship positions have systematically been moved 10 m to the west, while 10 m means each position has been moved 10 m to the east. The reference simulation (no uncertainty) is shown as a green solid line.

*Author contributions.* KK performed the analysis and wrote the manuscript with input from FW, AR, DB, AB, JPB, SF and OH. SF and OH were responsible for the measurements at the DURH and NERH measurement sites.

*Competing interests.* The authors declare that they have no conflict of interest.

*Acknowledgements.* The research presented in this study was funded in part by the EU Life project CLINSH and by the University of Bremen.

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
