# Peer review of "Determination of $NO_x$ emission rates of inland ships from on-shore measurements"

_EGUsphere, 2022_

## Author Comment (AC1)

**Comments from Fan Zhou:**

We would like to thank the reviewer for his helpful comments. We hope that we could address all questions and unclear points satisfactorily.

Legend: Author comments in blue, Referee comments in black.

The authors claim that "In contrast to relative emission factors (in grams per kilogram fuel), the emission rates (in grams per second) do not need further knowledge about the fuel consumption of the ship and can therefore be used directly to investigate the effect of ship traffic on air quality." In the part of Introduction, the relevant discussion is mainly about the measurement of emission factors. Whether there are other studies that measure emission rates? If there are relevant studies, it is recommended to supplement them and carry out necessary comparison (emission rates and emission factors), discussion, and analysis.

To our knowledge there are no studies which report on emission rates for inland ships. There are some studies reporting on $SO_2$ and $NO_2$ emission rates of sea ships (e.g., Berg et al., 2012; Berkhout et al., 2012; Prata, 2014). We added these to the introduction.

Emission factors are often used in the compilation of emission inventories, so can emission rates be used in the compilation of emission inventories? If so, whether there are relevant studies.

To our knowledge there are currently no studies which use emission rates for the compilation of an emission inventory. We think emission rates can be used for this task, but only for the region where they are derived.

In the introduction, it is suggested to supplement the discussion on the related research of inland ship emission monitoring, and the particularity of this research. On the whole, the content of the introduction is relatively small, so it is suggested that the authors make supplement on recommendations 1, 2, and 3.

We expanded the Introduction in regard to suggestions 1 and 2.

"The on-shore measurements were carried out using standardized air quality monitoring stations". I suggest a detailed introduction of the equipment, such as principle, accuracy, precision, measuring range, sensors. And comparison with related studies.

We added a detailed description of the used instruments, the measurement principle and the respective uncertainties to section 2.

I feel that the analysis of uncertainty factors is too little, and need to explain the possible error sources and effects in more detail.

We added more details regarding which uncertainty factors are important to section 3.5. We also added a Figure to the appendix to show an example of the Monte-Carlo-simulations (see

Figure 1). Specifically the following paragraph has been added:

*The largest sources of uncertainty of the derived emission rate are the wind speed, wind direction and stability. Wind speed and wind direction influence the shape and the time of appearance of the modelled peak. The area of the peak changes as a function of wind speed, lower wind speeds lead to a larger peak, while higher wind speeds lead to a smaller peak. Also the peaks shift in time. With lower wind speeds than assumed, the modelled plume arrives at the measurement site later than expected, while with higher than assumed wind speeds, it arrives too early. The wind direction has similar effects and also changes the peak area and the time of arrival of the peak maximum. Stability however only changes the modelled peak area, more unstable conditions lead to smaller modelled peaks, as the plume can also grow vertically and the pollutants are dispersed over a larger volume. In contrast more stable conditions leading to larger modelled peaks, as the vertical dispersion is hindered. The source position does not play such an important role and neither the changes in latitude, longitude or height show significant changes of the modelled peak within the considered uncertainties. Also the resulting uncertainty within the measured peak area is small compared to the uncertainty caused by the wind speed, wind direction and stability. An example of the Monte-Carlo-simulations and the respective influence on the modelled peaks can be found in the Appendix.*

[Figure]

**Figure 1:** Monte-Carlo simulations for the example plume shown in Figure 4. In each plot one parameter has been changed within its respective uncertainty and the resulting model peaks are shown. The uncertainties shown in the plot legends are always expressed as deviation from the reference simulation, e.g. in plot a) -10 m means that the ship positions have systematically been moved 10 m to the west, while 10 m means each position has been moved 10 m to the east. The reference simulation (no uncertainty) is shown as a green solid line.

6. In line 153, I think it would be clearer and more concise to present the results of the two experiments separately. Also, abbreviations do not seem to be used. DURH and NERH.

We split the results into parts, describing DURH and NERH separately. Also changed to use the abbreviations DURH and NERH.

7. If I understand correctly, this emission rate refers to the emission rate of the target ship (from AIS). Then I think it should be stated in the abstract and the text, otherwise there seems to be a certain ambiguity.

We've tried to adjust the summary and body to make things clearer.

8. Confusion of logic and structure in Result. The results of emission rate were chapter 4, compared results were chapter 4.1 and 4.2, respectively. Three subsections might be more appropriate; "In order to validate the emission factors within the CLINSH project", but the results in chapter 4 are emission rate. In other words, the result is emission rate, but validation is emission factor. Please clarify the logic in your argument.

"Emission factors" has been changed to "emission rates". We also refactored the whole Result section to improve readability.

**Technical corrections:**

1.Please add descriptions that $NO_x$=NO+$NO_2$, when the $NO_x$ first appeared.

Done.

2. Some of the symbols in Figure 2 are not clear, Va, Vb, IV, up, down.

Changed Figure 2 to show ship length instead of ship class, also the direction of travel was changed from "up" and "down" to "upstream" and "downstream".

3. line 154, Does "quality criteria" means that raised in 3.4 "Quality control"? If so, please mention it.

Done.

4. Figure 5, symbol don't know what it means. Although mentioned in Table 2, it seems inconvenient to read.

Thank you for your comment. Unfortunately, we could not find a way to make this figure more convenient to read.

5. In conclusion and other parts, one sentence as a paragraph is not recommended unless it's an important conclusion.

Changed all one sentence paragraphs to include them into previous / following paragraphs.

**References**

Berg, N., Mellqvist, J., Jalkanen, J.-P., and Balzani, J.: Ship emissions of $SO_2$ and $NO_2$: DOAS measurements from airborne platforms, Atmospheric Measurement Techniques, 5, 1085–1098, https://doi.org/10.5194/amt-5-1085-2012, 2012.

Berkhout, A. J. C., Swart, D. P. J., van der Hoff, G. R., and Bergwerff, J. B.: Sulphur dioxide emissions of oceangoing vessels measured remotely with Lidar: RIVM Report 609021119/2012, 2012.

Prata, A. J.: Measuring SO$_2$ ship emissions with an ultraviolet imaging camera, Atmospheric Measurement Techniques, 7, 1213–1229, https://doi.org/10.5194/amt-7-1213-2014, 2014.

---

## Author Comment (AC2)

**Comments from anonymous Referee #2:**

We would like to thank the reviewer for his/her helpful comments. We hope that we could address all questions and unclear points satisfactorily.

Legend: Author comments in blue, Referee comments in black.

**General comments:**

I have the feeling that the method described here is very similar to the one presented in a former paper by Krause et al. (2021), except that the concentration at one specific point (in-situ measurement) rather than the mean concentration along the light path (LP-DOAS) is considered in the Gaussian puff model. As the description of the method is a substantial fraction of the manuscript, the authors should highlight what is new about it and what the major improvements are.

We tried to highlight the improvements compared to Krause et al. (2021) in the text. Specifically, we added the following to section 3.3.:

*In contrast to Krause et al. (2021), the whole ship passage is modelled which allows to model the transport and dispersion of the emitted trace gases as a function of time. Consequently, the peak area can be used to derive the respective emission rate of the ship, whereas in Krause et al. (2021) only the peak maximum was identified and compared with the modelled maximum to derive the emission rate. In comparison, using the peak area is a more reliable measure than the peak maximum, because it relates the total amount of pollutants arriving at the measurement site.*

I also think that the uncertainty of the method should be discussed in more detail. This can be done in the supplement by showing examples of the Monte-Carlo-simulations and providing uncertainty estimates. See also specific comments below.

We added more details regarding which uncertainty factors are important to section 3.5. We also added Figure 1 showing an example of the Monte-Carlo-simulations in the appendix.

The structure of the results section is a bit confusing. For the reader it is not clear at what point results from which station are presented. If I understand it correctly, most of the text as well as Fig. 5-8 are only about DURH. This should be pointed out. The authors should consider splitting it into two subsections DURH (ships traveling) and NERH (ships entering/leaving harbor) and potentially extent the conclusions drawn from the study at NERH which seems to be underrepresented (1 sentence and Fig. 9).

We split the results into two sections, one for DURH and one for NERH.

The authors state that emission rates in grams per second are favored and have some advantages (e.g. no assumption about fuel consumption needed) over emission rates in grams per kilogram

fuel (p.1, l. 11-13 and p.2, l. 45-47). I think this should be explained in more detail in the introduction, as most of the literature reports emission rates in g per kg or g per kWh. What are applications (model simulations, emission inventories etc.) the results can be used for and what are potential limitations on the other hand?

*We tried to improve on this part of the introduction and added some information about possible applications.*

**Specific comments:**

•p.1, l. 11-13: Here it is implied that for the derivation of emission rates in g per kg fuel the knowledge about the fuel consumption is required. However, it could be calculated from measured NOx to CO2 ratio without this knowledge. Only for the transformation from g per kg fuel into g per s (or vice versa) the fuel consumption would be needed.

*We rephrased l. 11-13 to make things clearer.*

*The emission rates (in grams per second) can be directly used to investigate the effect of ship traffic on air quality, as the absolute emitted amount of pollutants per unit time is known. In contrast, for relative emission factors (in grams per kilogram fuel), further knowledge about the fuel consumption of the individual ships is needed, to calculate the amount of pollutants emitted per unit time.*

•p.1, l.16: Talking about inland vessels explicitly, I would not consider SO2 as a significant source of emissions as the sulfur content in the diesel fuel is very low.

*We agree and removed $SO_2$ from the sentence.*

•p.2, l.53: What instrument was used? Specifications?

*We added a detailed description of the used instruments, the measurement principle and the respective uncertainties to section 2.*

•p.5, l.87: What was done when the atmospheric variability was high and the threshold of 2 ppbv was exceeded due to other point sources?

*This is not explicitly corrected for. The station generally shows a slow fluctuation of the background $NO_x$ and ships seem to be the only strong point sources that cause peaks in $NO_x$.*

•p.5, l.112: How is the funnel height estimated and what value is assumed in the model?

*We added the following information to section 3.3.:*

*The funnel height is assumed to be 5 m above the mean water level, which was always assumed to be at the surface level (z=0). The height of 5 m was chosen, because it is assumed, that the plume quickly bends down due to wind and movement of the ship. The height was always assumed to be the same, as most inland ships share a similar distinctive form, with the funnel*

*at the back of the ship in similar heights.*

•p.6, l.140 "If the Monte-Carlo-simulations and the reference simulation do not show large deviations, the derived $NO_x$ emission rates for that specific case are used for further evaluation": What does "large" mean in this context? Is a reasonable deviation more like 10 % or 50 %? I would especially be interested in the importance of the assumed plume height, as water level and funnel height are estimated and will change over the course of the year respectively differ from ship to ship.

A reasonable deviation is defined to be below 50 % of the reference value. Figure 1 (added to the appendix of the paper) shows the results of the Monte-Carlo-simulations for one ship passage. The height of the plume does not seem to be important compared to the influence of the stability, the wind speed and the wind direction.

[Figure]

**Figure 1:** Monte-Carlo simulations for the example plume shown in Figure 4. In each plot one parameter has been changed within its respective uncertainty and the resulting model peaks are shown. The uncertainties shown in the plot legends are always expressed as deviation from the reference simulation, e.g. in plot a) -10 m means that the ship positions have systematically been moved 10 m to the west, while 10 m means each position has been moved 10 m to the east. The reference simulation (no uncertainty) is shown as a green solid line.

•p.7, l.151: The calculation of uncertainties is presented in detail but quantitative values are not given anywhere. Although titled as "negligible", I suggest presenting them in the supplement.

We added the individual $NO_x$ emission rates derived from on-shore measurements to the supplement.

•p.7, l.153: What was the percentage of ship peaks identified relative to the total number of ship passages?

At DURH there were approximately 291000 ship passages (derived from AIS). In total 32910 peaks were identified and could be attributed to these ship passages, of which about 23600 passed the quality criteria.

•p.9, l.177 "Ships that are not influenced by the current show similar NO$_x$ emission rates independent of direction of travel (e.g. Figure 9).": It is not clear at first sight that this refers to ships leaving or entering the harbour at NERH (see general comment above).

We refactored the results section and split the results of DURH and NERH so it should be more obvious which figure shows the results of which measurement station.

• p.10, Fig. 5: The differences between ship classes and potential reasons should be further discussed in the text.

We expanded this section of the text with more information. Specifically the following paragraph has been edited:

*The NO$_x$ emission rates in the context of size (or ship class) are more difficult to summarize (see Figure 5). Generally, larger ships show larger NO$_x$ emission rates than smaller ships. Larger ships usually have more powerful engines to provide the power needed to move and manoeuvre the ship. More powerful and larger engines consume more fuel and therefore have higher emission rates than smaller engines. At the same time, the larger ships are usually newer and their emissions are regulated, while older ships are subject to grandfathering, which means their engines do not have to comply with new regulations. Only if the engine of an older ship is exchanged, are the new regulations applicable. Due to the long service life of inland ships, a lot of the smaller ships do not fall under the regulations and therefore still have high emissions.*

• p.10, l.179: What was the ratio of identified ship peaks for ships traveling upstream versus downstream?

Of the 23600 quality checked emission rates derived at DURH, about 3500 are categorized as downstream direction, 13600 as upstream direction and 6500 do not fall in either category, because the ships changed their course throughout the considered time frame, e.g. to enter or leave DURH or one of the connecting channels.

• p.11, l.196: "as the uncertainty of the Gaussian-puff model is quite high": What does quite high mean (see above)?

A more detailed description of the uncertainties of the Gaussian-puff model has been added to section 3.5. Specifically the following paragraph:

*The largest sources of uncertainty of the derived emission rate are the wind speed, wind direction and stability. Wind speed and wind direction influence the shape and the time of appearance of the modelled peak. The area of the peak changes as a function of wind speed, lower wind speeds lead to a larger peak, while higher wind speeds lead to a smaller peak. Also the peaks shift in time. With lower wind speeds than assumed, the modelled plume arrives at the measurement site later than expected, while with higher than assumed wind speeds, it arrives too*

*early. The wind direction has similar effects and also changes the peak area and the time of arrival of the peak maximum. Stability however only changes the modelled peak area, more unstable conditions lead to smaller modelled peaks, as the plume can also grow vertically and the pollutants are dispersed over a larger volume. In contrast more stable conditions leading to larger modelled peaks, as the vertical dispersion is hindered. The source position does not play such an important role and neither the changes in latitude, longitude or height show significant changes of the modelled peak within the considered uncertainties. Also the resulting uncertainty within the measured peak area is small compared to the uncertainty caused by the wind speed, wind direction and stability. An example of the Monte-Carlo-simulations and the respective influence on the modelled peaks can be found in the Appendix.*

A more detailed description of the quality control has been added to section 3.4. Specifcally:

*The assumptions made in the model to estimate the $NO_x$ emission rate may not truly represent the conditions at the time of measurement. To assess the quality of the derived emission rate, Monte-Carlo-simulations are performed to assess whether a small change in one of the input parameters results in a large change of the derived concentration at the measurement site. The parameters varied are wind speed, wind direction, atmospheric stability and the position of the ship in longitude, latitude and height. Each of these parameters is changed within the uncertainty ranges given in Table 1 (an example can be found in Appendix C). For each changed parameter, the derived integrated peak concentrations are then compared to the integrated peak concentration of the reference simulation. The resulting concentrations of a set of Monte-Carlo-simulations are then summarized by the mean value ($mean_{Cj}$), the standard deviation ($\sigma_{Cj}$) and the minimum ($min_{Cj}$) and maximum value ($max_{Cj}$). These values are compared to the reference simulation of the unperturbed input parameters. To be evaluated further, the following five criteria must be met by the set of Monte-Carlo-Simulations for each input parameter:*

*1) $mean_{Cj} / C_{model}$ must be between 0.5 and 1.5, to eliminate cases with a systematic deviation caused by the uncertainty of a single input.*

*2) $\sigma_{Cj} / C_{model}$ must be lower or equal to 1, to eliminate cases with a high variability caused by the uncertainty of a single input.*

*3) The difference between $min_{Cj} / C_{model}$ and $max_{Cj} / C_{model}$ must be smaller than 2, to eliminate cases with a large spread between minimum and maximum of the set due to the uncertainty of a single input.*

*4) the absolute error of the derived emission rate must be lower than $5 \, \mathrm{g \, s^{-1}}$, which eliminates cases, where the uncertainty is on a larger order of magnitude than the emission rate.*

*5) the relative error of the derived emission rate must be smaller than 200 %, which eliminates cases, where the uncertainty is much larger than the emission rate.*

• p.12, l.217-l.219: What water velocity and average speeds for ships traveling upstream and downstream are these specific fuel consumptions reported by Allekotte et al. based on? For me the difference between 108 kg per h and 162 kg per h for the low/high fuel consumption scenarios seems quite large, given an average ship speed over water of 3 m/s and 5 m/s at DURH station and assuming a water velocity of 1 m/s . Please clarify.

The energy consumptions are based on model calculations described in Knörr et al. (2013), which further rely on Bundesministerium für Verkehr (2011). Ship speed over ground is 2.8 m/s for ships travelling downstream and 2.5 m/s for ships travelling upstream (Knörr et al., 2013). Water velocity of the Rhine is considered to be 1.7 m/s (Knörr et al., 2013). Allekotte et al. (2020) improved these with information of on-board measurements of different vessels presented in van Mensch et al. (2018). The Bundesministerium für Verkehr (2011) values are based on an economic operation mode, whereas beforehand Knörr et al. (2013) assumed that ships always want to achieve maximum speed. The values presented in (Allekotte et al., 2020) are at the upper limit of the Bundesministerium für Verkehr (2011) values, but below the values of (Knörr et al., 2013).

• p.13, l. 226 "rates fit into the range" and Table 4: For ships traveling downstream the deviation is much higher than for ships traveling upstream. The authors should discuss potential reasons for this (see above).

From the gathered data we assume that ships travel with the same speed in water, independent of direction of travel. Further more assuming a water velocity of about 1 m/s, the average speed in water would be similar for both directions, and thus lead to similar engine operation conditions. This is confirmed by the similar average emission rates at the average speed over ground in both direction. We also the following paragraph to the text:

*At the most common speeds over ground in a given direction, the emission rates for ships travelling upstream and ships travelling downstream are similar. In general, the scenario with high fuel consumption seems to better reflect the derived emission rates. This indicates similar fuel consumption and engine operation scenarios for ships travelling downstream and ships travelling upstream. Assuming a water velocity of $1 \mathrm{~m} \mathrm{~s}^{-1}$, the average speed in water would be similar for both directions, which would also indicate similar engine operation conditions and therefore similar emission rates.*

• p.13, l.229 "using a specific fuel consumption of 230 g per kWh": The uncertainty of the specific fuel consumption which is often used in literature should definitely be discussed at some point in the manuscript, since the authors say that their method has the advantage of not requiring the fuel consumption and the fuel consumption is used to convert their results to compare them with CCNR regulations.

We added some information about the specific fuel consumption:

*The specific fuel consumption of ship engines can vary between different engines and also depends on the engine load, lower engine load generally increasing the specific fuel consumption (van Mensch et al., 2018). Measurements presented by van Mensch et al. (2018) show that the specific fuel consumption for different engines can reach about 290 $\mathrm{g\,kWh^{-1}}$ for engine loads below 20 % and between 200 and 230 $\mathrm{g\,kWh^{-1}}$ for engine loads higher than 20 %. Additionally, the specific fuel consumption also depends on the age of the engine, newer engines generally having a lower specific fuel consumption (De Vlieger et al., 2004).*

• p.15, Table 3: Can you say anything about the typical motor operating conditions (e.g. engine load/rpm) during the on-board measurements? Did you maybe observe any difference in emission rates when varying these conditions?

The emission rates change as a function of engine rpm. An example for a measurement is shown in Figure 2.

[Figure]

**Figure 2:** Example of on-board measurements made on a ship participating in the CLINSH project.

• p.18, l.273: Have there been any point sources of NOx next to DURH or NERH that might also cause peak-like structures and if yes, how did you make sure to exclude them from the analysis?

There is a chemical company southwest of the DURH measurement station, which emitted about 244000 kg of $NO_x$ in 2018 (translates to roughly 7.7 g/s). It is not reported where exactly emissions take place, but aerial pictures show several chimneys, also chimneys are higher than surroundings. We guess this only influences the background, as it is a constant source which should arrive at the measurement station well mixed with the ambient air (2.5 km distance) and only for specific wind directions (South westerly winds). Also, the amount of emitted $NO_x$ reduced over time (145000 kg in 2019 and zero reported $NO_x$ emissions in the following years)

(Umweltbundesamt, 2022). Additionally, there is a highway bridge across the river Rhine also located southwest of the DURH measurement site. However, the amount of traffic across the bridge is limited to two lanes in total. Individual cars or trucks should not cause strong enhancements such as the passing ships and total emission caused should only influence the background $NO_x$ concentration. Closer possible source are obstructed by buildings and generally align with wind directions that can not be evaluated for ship plumes (North easterly to south easterly winds).

At NERH there is a highway bridge close to the measurement site, but again individual cars or trucks should not cause strong peaks and only influence the background $NO_x$ concentration.

• p.20, l.313-314: The authors should discuss to what extent the emission rates derived for Duisburg would be transferable to other locations along the Rhine.

Generally, the derived emission rates should be representable for the lower Rhine area, with similar streaming conditions as they are encountered at the DURH measurement site. Added the following to the conclusions section:

*Generally, the derived emission rates should be representative for the lower Rhine area, with similar streaming conditions as they are encountered at the DURH measurement site. At the same time, evaluation of the AIS signals derived from DURH and NERH show, that ships tend to adapt their speed to streaming conditions encountered at each measurement site, which could also influence their emission rates.*

**Technical corrections:**

• Title: I would suggest to omit the word "sailing", it could be confused with ships using sails.

Thank your for your suggestion, we will omit "sailing" in the title.

• p.2, l.27: [. . . ] which derived emission rates [. . . ]

Thank your for your suggestion. The whole paragraph was rewritten to include comments suggested by another reviewer.

• p.7, l.145: In Eq. 3 it should be dQmodel in the second term?

We checked the formula and think it is correct. $Q_{model}$ only is a fixed arbitrary number which has no uncertainty.

• p.8, l.162: [. . . ] majority of ships [. . . ]

Done.

• p.10, Fig. 5: Due to the large number of dots representing single measurements and the chosen range, for me it is hard to compare e.g. the median values of each ship class with each other. I would suggest adding a table in the supplement with corresponding statistics (mean, median,

range).

We added the respective results as tables to the appendix.

• p.15, Table 3: on-board median is missing.

Since the median values are close to the respective mean values, we omit them for clarity.

• p.20, l.316: [. . . ] emission rates [. . . ]

Changed emission factors to emission rate.

**References**

Allekotte, M., Biemann, K., Heidt, C., Colson, M., and Knörr, W.: Aktualisierung der Modelle TREMOD/TREMOD-MM für die Emissionsberichterstattung 2020 (Berichtsperiode 1990-2018), 2020.

Bundesministerium für Verkehr, Bau-und Wohnungswesen, U. W.: Nutzen-Kosten-Analysen (NKA) für Investitionen an Binnenschifffahrtsstraßen: Kompendium, Bundesministerium für Verkehr, Bau und Stadtentwicklung, Unterabt. Wasserstraßen, Berlin, 2011.

De Vlieger, I., Int Panis, L., Joul, H., and Cornelis, E.: Fuel consumption and $CO_2$-rates for inland vessels, Urban Transport X, https://doi.org/10.2495/UT040621, 2004.

Knörr, W., Heidt, C., Schmied, M., and Notter, B.: Aktualisierung der Emissionsberechnung für die Binnenschifffahrt und Übertragung der Daten in TREMOD, 2013.

Krause, K., Wittrock, F., Richter, A., Schmitt, S., Pöhler, D., Weigelt, A., and Burrows, J. P.: Estimation of ship emission rates at a major shipping lane by long-path DOAS measurements, Atmospheric Measurement Techniques, 14, 5791–5807, https://doi.org/10.5194/amt-14-5791-2021, 2021.

Umweltbundesamt: URL `https://thru.de/thrude/`, 2022.

van Mensch, P., Abma, D., Verbeek, R., and Hekman, W.: PROMINENT Report D5.7 Technical evaluation of procedures for Certification, Monitoring and Enforcement. Technical evaluation of the monitoring results on Rhine, Danube and other vessels. Public report version 2, 2018., 2018.